# Influence of Biomimetically Mineralized Collagen Scaffolds on Bone Cell Proliferation and Immune Activation

**DOI:** 10.3390/polym14030602

**Published:** 2022-02-03

**Authors:** Lucie Bacakova, Katarina Novotna, Daniel Hadraba, Jana Musilkova, Petr Slepicka, Milos Beran

**Affiliations:** 1Institute of Physiology of the Czech Academy of Sciences, Videnska 1083, 142 20 Prague 4, Czech Republic; knovotna@gmail.com (K.N.); Daniel.Hadraba@fgu.cas.cz (D.H.); Jana.Musilkova@fgu.cas.cz (J.M.); 2Department of Solid State Engineering, Faculty of Chemical Technology, University of Chemistry and Technology, Technicka 5, 166 28 Prague 6, Czech Republic; Petr.Slepicka@vscht.cz; 3Food Research Institute Prague, Radiova 7, 102 31 Prague 10, Czech Republic; Milos.Beran@vupp.cz

**Keywords:** biopolymers, nature-derived polymers, porous scaffolds, mechanical properties, elemental composition, regenerative medicine, cell adhesion, cell growth, cell differentiation, inflammatory activation

## Abstract

Collagen, as the main component of connective tissue, is frequently used in various tissue engineering applications. In this study, porous sponge-like collagen scaffolds were prepared by freeze-drying and were then mineralized in a simulated body fluid. The mechanical stability was similar in both types of scaffolds, but the mineralized scaffolds (MCS) contained significantly more calcium, magnesium and phosphorus than the unmineralized scaffolds (UCS). Although the MCS contained a lower percentage (~32.5%) of pores suitable for cell ingrowth (113–357 μm in diameter) than the UCS (~70%), the number of human-osteoblast-like MG-63 cells on days 1, 3 and 7 after seeding was higher on MCS than on UCS, and the cells penetrated deeper into the MCS. The cell growth in extracts prepared by eluting the scaffolds for 7 days in a cell culture medium was also markedly higher in the MCS extracts, as indicated by real-time monitoring in the sensory xCELLigence system for 7 days. From this point of view, MCS are more promising for bone tissue engineering than UCS. However, MCS evoked a more pronounced inflammatory response than UCS, as indicated by the production of tumor necrosis factor-alpha (TNF-α) in macrophage-like RAW 264.7 cells in cultures on these scaffolds.

## 1. Introduction

Synthetic and nature-derived polymers are widely used as permanent or temporary scaffolds for cells in hard and soft tissue engineering applications. Both groups of polymers include materials which are either stable or degradable in the human organism. Examples of synthetic stable polymers include expanded polytetrafluoroethylene (ePTFE) and polyethylene terephthalate (PET), which are applied, e.g., for the fabrication of clinically used vascular prostheses [1] and also for bone tissue regeneration [2,3]. More advanced tissue-engineered replacements, which are generally still under experimental development, are made of degradable synthetic polymers, such as polylactic acid, polyglycolic acid, polycaprolactone and their copolymers [1,4]. Nature-derived polymers that are non-degradable or slowly degradable after implantation into the human organism include polysaccharides or proteins synthesized by bacteria, fungi, algae, plants or animals, such as cellulose, chitosan or silk fibroin. Finally, nature-derived polymers that are easily degradable in the human organism include biopolymers that occur in the extracellular matrix of human and other mammalian tissues, such as collagen, elastin, fibronectin and laminin (for a review, see [5]).

This study is focused on three-dimensional (3D) porous scaffolds based on collagen for bone tissue engineering. Porous scaffolds are an important type of 3D materials used for tissue engineering, especially for bone tissue engineering, because they resemble the physiological architecture of native bone tissue. The void spaces in these scaffolds allow the ingrowth and accommodation of osteoblasts, diffusion of nutrients and removal of waste products, production of mineralized bone matrix by osteoblasts, vascularization of the newly formed bone tissue, and also the formation of osteon-like structures [6,7,8,9]. Collagen is a major protein component of the extracellular matrix of bone tissue. It is a biopolymer of amino acids (mainly glycine, proline, alanine, hydroxyproline), characterized by its primary, secondary and tertiary structure (for a review, see [10,11]). Collagen enables the direct adhesion of cells to specific amino acid sequences within its molecules, such as GFOGER in physiological native triple-helical collagen [12] and DGEA and RGD in modified collagen, such as heat-denatured collagen and gelatin [13]. In addition, collagen is biodegradable and reorganizable by cells and can be easily replaced with their own newly synthesized extracellular matrix. Last but not least, collagen provides the structural matrix upon which mineralization occurs [10,11].

However, like most natural or synthetic polymers, collagen itself has insufficient mechanical properties for hard tissue engineering, particularly in parameters obtained from a stress–strain curve, such as Young’s modulus of elasticity (*E*). In human bone tissue, *E* has been reported to range approx. from 2 to 22 GPa depending on the evaluation method, dry or wet conditions during evaluation, the anatomical site or the age and gender of the donor (for a review, see [14]). The *E* of bovine tendon collagen was relatively high, ranging from 1.0 to 3.9 GPa [15], and for individual type I collagen fibrils from a rat tail, *E* even ranged from 5 to 11.5 GPa [16]. However, the *E* of porous collagen scaffolds used for bone tissue engineering was generally much lower—for example, the *E* of collagen scaffolds prepared by freeze-drying only ranged from 1 to 10 kPa depending on whether they were in a hydrated or dry state [17]. Thus, the porous collagen scaffolds designed for bone tissue engineering need some reinforcement. In native bone tissue, collagen occurs in combination with a mineral component, formed by calcium phosphates, mainly in an apatite form (for a review, see [10,11]). Tissue engineers have therefore often used a combination of collagen with similar inorganic materials, such as hydroxyapatite (HAp) [18,19,20], tricalcium phosphate [21] or bioactive glass [22].

The presence of a mineral component not only improves the mechanical properties of the scaffolds but can also enhance their interaction with cells via mechanical and/or biochemical signaling. For example, collagen–agarose blends tuned to the optimum stiffness by the addition of tricalcium phosphate microparticles increased the osteogenic differentiation of human bone marrow mesenchymal stem cells, manifested by an increased expression of genes for runt-related transcription factor 2 (RUNX2), type I collagen and alkaline phosphatase [21]. Dissolution products from composite bioglass–carbonate apatite–collagen scaffolds, containing calcium ions, soluble silicon species and phosphorus, promoted osteogenic differentiation in primary human osteoblasts, manifested by an increased expression of genes for osteopontin and osteocalcin [22]. Mineral particles within collagen-based scaffolds can also be utilized as carriers for the controlled delivery of various biologically active molecules, such as anti-osteoporotic drugs [18], bone morphogenetic protein-2 [19] and vascular endothelial growth factor [20].

The mineral component has been incorporated into the collagen scaffolds by two main procedures, i.e., by admixing mineral micro- or nanoparticles into the collagen matrix [8,9,10,11,12] or by a spontaneous process referred to as biomimetic mineralization. Biomimetic mineralization includes “classical” incubation of the scaffolds in a simulated body fluid (SBF) or in other related fluids containing calcium, phosphate and other ions [9,23] or a more sophisticated, bioinspired polymer-induced liquid precursor (PILP) process, in which the mineralization of collagen in ionic solutions is facilitated by the presence of various anionic polymers that mimic negatively charged non-collagenous proteins in the native bone tissue [24]; for a review, see [10,11,25]. Biomimetic mineralization of the collagen matrix can also be realized directly by cells, e.g., by dental pulp stem cells [26] or even by osteosarcoma-derived line cells, such as osteoblast-like Saos-2 cells [27]. Another advanced approach is enzyme-assisted biomineralization, based on the incorporation of alkaline phosphatase, an enzyme involved in bone matrix mineralization [25], or phytase, a plant-derived phosphatase [28], into collagen and other hydrogel scaffolds.

Spontaneous biomineralization of collagen scaffolds mimics the physiological process of bioapatite accommodation by collagen better than artificial procedures in which mineral particles are admixed into a collagen matrix. The chemical composition of the mineral component in biomineralized scaffolds, its interaction with the collagen component, the ratio between the inorganic and organic components and the mechanical properties of these scaffolds are in general more similar to those of native bone tissue. The osteoconductive and osteoinductive performance of biomineralized scaffolds is therefore better than in scaffolds with artificially admixed mineral fillers (for a review, see [29,30,31]). Biomineralized collagen can be processed by advanced techniques, such as 3D printing together with cells [10,31], and can be used for gene delivery [32] and for osteogenic differentiation of stem cells [33].

An important question, which has been relatively rarely taken into account, is the potential immunogenicity of collagen-based scaffolds. Bovine and porcine collagens, the most widely used collagens in tissue engineering, have often been reported as immunogenic and evoking an inflammatory response [34,35,36]. Even non-mammalian collagens, derived, e.g., from jellyfish [37] or from fish [38], which are highly popular in modern tissue engineering, can elicit a considerable immune response, often comparable with that of mammalian collagen [38]. Interestingly, the immunogenicity of collagen can be reduced by biomimetic mineralization [39,40].

In the study presented here, porous collagen scaffolds were therefore modified by biomimetic mineralization in SBF and were characterized from the point of view of their morphology, elemental composition, structural changes and mechanical properties. The adhesion and growth of human-osteoblast-like MG-63 cells were studied in cultures on these scaffolds and were also monitored in extracts from these scaffolds, using a Real-Time Cell Analyser (xCELLigence), based on an impedance sensor. The osteogenic cell differentiation was evaluated by an enzyme-linked immunosorbent assay (ELISA) of osteocalcin. An investigation was also carried out on the immune response, as measured by the production of TNF-α, a pro-inflammatory cytokine, by rat-macrophage-like RAW 264.7 cells cultured in the presence of the scaffolds. We found that mineralized scaffolds provided more suitable support for colonization with MG-63 cells than unmineralized scaffolds but induced greater production of TNF-α in RAW 264.7 cells.

## 2. Materials and Methods

### 2.1. Preparation and Mineralization of the Scaffolds

#### 2.1.1. Preparation of Collagen Scaffolds

A collagen suspension was prepared by gradual addition of bovine collagen I (MG, Medicol, Prague, Czech Republic) to 0.5 l of demineralized water (DM), containing 0.2 mL of ammonia, under continuous stirring to obtain a resulting collagen concentration of 1% (*w*/*v*) of DM and pH 6.5. The suspension was intensively stirred for an additional 30 min. Then, the suspension was whipped in a high-speed mixer, poured into dishes, frozen immediately and freeze-dried to obtain a three-dimensional porous structure. The sponge material was cut into 10 × 10 mm or 15 × 15 mm square-shaped pieces 1 mm in height.

#### 2.1.2. Biomimetic Mineralization of Collagen Scaffolds with SBF

A modified SBF (m-SBF) was prepared according to a published protocol [41] in the 1M Tris–HCl buffer, pH 7.4, with concentration of ions (Na^+^, K^+^, Mg^2+^, Ca^2+^, Cl^−^, HCO_3_^−^, HPO_4_^2−^ and SO_4_^2−^) equal to or close to those of blood plasma. The collagen sponges were immersed into the m-SBF. Two hundred ml of the m-SBF solution was used for each gram of the freeze-dried collagen sponge. The sponges were incubated in the solution at 36.5 °C for 48 h under slow stirring. After 24 h, the solution was changed for fresh one. Finally, the sponges were rinsed repeatedly with excess of demineralized water to remove residual inorganic ions, frozen immediately and freeze-dried.

### 2.2. Characterization of the Scaffolds

#### 2.2.1. Morphology of the Scaffolds

The morphology of both the unmineralized collagen scaffolds (UCS) and the mineralized collagen scaffolds (MCS) was evaluated by scanning confocal microscopy. Both the UCS and the MCS were measured in two states—dry and wet. The wet condition better mimics the cell cultivation environment, and the samples were therefore immersed in a physiological solution for 3 days. After that, the dry and wet samples were optically sectioned by an upright scanning confocal microscope (Leica TCS SPE DM2500; Leica, Wetzlar, Germany), obj. ACS APO 10x/0.30 CS and C PLAN 4x/0.10. Optical sections were acquired from the top surface structures to a depth of 490 µm with a step size of approx. 22 µm. The samples were excited at 405 nm, and the autofluorescence signal was detected at 450 to 600 nm. The size of the pore entrances (entering pore area) was measured manually using Fiji software on microphotographs taken in 8 randomly selected regions for each type of scaffold. In each picture, 15 randomly selected pores were measured (i.e., 120 pores in total for each type of scaffold).

In addition, the morphology of the scaffolds was evaluated by scanning electron microscopy (SEM). The samples of the scaffolds were fixed by a double-faced adhesive tape to the holders and were evaluated in a Phenom G2 Scanning Electron Microscope (Phenom-World BV, Eindhoven, The Netherlands).

#### 2.2.2. Evaluation of Texture Stability under a Mechanical Load

A TA-XT2i texture analyser (Stable Micro Systems, Godalming, UK) was used for an evaluation of the texture stability of the scaffolds by conducting tensile and compression tests (Figure 1). The ultimate tensile strength at the moment when the sample ruptured was measured using three strips with the same geometry (90 mm in length and 30 mm in width). The strips were fixed into the jaws, which were dilated at a rate of 0.1 mm/s until the strips were ruptured. The measurement was repeated 3 times. The material compression strength was measured using a round probe with a cross section of 0.5 cm^2^. The speed of the probe was 0.5 mm/s. For each type of scaffold, 10 measurements were performed at 75% relative deformation.

#### 2.2.3. Elemental Composition of the Scaffolds

Elements were extracted from the samples by homogenizing them with acids and hydrogen peroxide prior to analysis. The elements (Ca, K, Mg, Na, P) were quantified by inductively coupled plasma atomic emission spectrometry (ICP-OES) by ALS Czech Republic, s.r.o., Prague, Czech Republic.

#### 2.2.4. Structural Changes in the Scaffolds after Mineralization

The structural changes in the scaffolds after mineralization in SBF were evaluated by an infrared spectroscopy analysis, performed using a Fourier-transform infrared (FTIR) spectrometer Nicolet iS5 (Fisher Scientific, Waltham, MA, USA) with a diamond crystal iD7 ATR accessory. The spectra were obtained as an average from 128 measurement cycles in the spectral range of 4000–550 cm^−1^ with the data interval of 0.964 cm^−1^. An atmospheric suppression feature was employed to eliminate ambient CO_2_ and H_2_O concentration changes.

### 2.3. Cell Model and Culture Conditions

UCS and MCS were sterilized by γ-irradiation and were inserted into 12-well or 24-well polystyrene multidishes (TPP, Trasadingen, Switzerland). In order to prevent them floating in the medium, they were fixed to the bottoms of the wells with CellCrown™ Cell Culture Inserts (Scaffdex Oy, Tampere, Finland). The scaffolds were then seeded with human osteoblast-like MG-63 cells (European Collection of Authenticated Cell Cultures, Porton Down, Salisbury, UK; Cat. No. 86051601). These cells were chosen because they are considered to provide a suitable model for studies on adhesion, expression of integrin subunits, proliferation, mechanosensitivity and production of osteocalcin by human-osteoblast-like cells [42,43].

The cells on the materials were cultured in Dulbecco’s modified Eagle’s Minimum Essential Medium (DMEM; Sigma, St. Louis, MO, USA, Cat. No. D5648) supplemented with 10% of fetal bovine serum (FBS; Sigma-Aldrich, St. Louis, MO, USA, Cat. No. F7524) and gentamicin (40 µg/mL, LEK, Ljubljana, Slovenia). Each 24-well dish contained a sample 10 × 10 × 1 mm in size, 30,000 cells and 1.5 mL of medium (for PicoGreen dsDNA assay, cell visualization and microscopy and an evaluation of the potential cell immune activation), and each 12-well dish contained a sample 15 × 15 × 1 mm in size, 100,000 cells and 2.5 mL of medium (for ELISA). The seeding density for real-time monitoring in the xCELLigence system was 3000 cells/well in 200 μL of the extract medium.

The cells were cultured for 1, 3 and 7 days at 37 °C in a humidified atmosphere containing 5% of CO_2_ in the air. Pure polystyrene wells (PS) or microscopic glass coverslips were used as reference samples.

### 2.4. Cell Adhesion, Growth and Viability

#### 2.4.1. Visualization of Cells within the Scaffolds

On day 3 and 7 after seeding, the cells were rinsed with phosphate-buffered saline (PBS; Sigma, St. Louis, MO, USA), were fixed with 70% ethanol (−20^·^°C, 5 min) and were stained with propidium iodide. The distribution and the shape of the cells within the scaffolds were evaluated using an inverted confocal microscope (Leica TCS SP2, Wetzlar, Germany). The cell ingrowth inside the materials was evaluated by summarizing the horizontal optical sections recorded every 10 μm for the scaffolds and every 3 μm for the glass coverslips. The actual depth of the cells within the scaffolds was indicated by color-coded projection.

#### 2.4.2. PicoGreen dsDNA Assay

The cell adhesion and growth were estimated by measuring the quantity of DNA in the scaffolds on days 1, 3 and 7 after seeding. The scaffolds with cells were rinsed with PBS and were incubated in 700 μL of lysis buffer (0.05% sodium dodecyl sulfate) for 15 min on the plate shaker. The quantity of double-stranded DNA in the solution was measured by the Quant-iT™ PicoGreen^®^ dsDNA kit (Invitrogen, ThermoFisher Scientific, Waltham, MA, USA) and by the fluorescence microplate reader (Synergy HT Multi-Mode Reader, BioTek, Winooski, VT, USA), according to the manufacturer’s protocol. Three independent samples for each experimental group and time interval were used.

#### 2.4.3. Real-Time Monitoring of Cell Growth in Extracts from the Scaffolds

UCS and MCS were eluted in DMEM with FBS for 7 days, and these eluates were used as media for cell growth, which was recorded by the xCELLigence Real-Time Cell Analyser (Roche, Basel, Switzerland). The cells were cultured in special 96-well E-plates with gold micro-electrodes incorporated into the bottoms of the wells. The system is based on measuring the electrical impedance, which is presented as the cell index. The cell index correlates positively with the coverage of the well bottoms with cells, i.e., with the cell number and spreading, and was measured every 15 min for the first 24 h and then every 30 min for the following 6 days.

### 2.5. Molecular Markers of Cell Adhesion, Spreading and Osteogenic Differentiation

The concentration of vinculin, a protein of focal adhesion plaques, β-actin, a cytoskeletal protein and osteocalcin, a calcium-binding non-collagenous protein of the bone extracellular matrix, was evaluated by an enzyme-linked immunosorbent assay (ELISA). The scaffolds with MG-63 cells were rinsed with PBS and were inserted into 500 μL of lysis buffer (0.05% sodium dodecyl sulfate, SDS) for 15 min and were kept on the plate shaker, and the total protein content was measured using a Coomassie Plus (Bradford) Kit (ThermoFisher Scientific, Rockford, IL, USA, Cat. No. 23236). The cell lysates were diluted to the same concentrations of total protein (100µg/mL). Aliquots of the cell lysates in various concentrations (corresponding to 1–50 μg of protein in 50 μL of water) were adsorbed on 96-well microtiter plates (Nunc-Immuno™ Plates, F96 MaxiSorp, Cat. No. 442404, Roskilde, Denmark) at 4 °C overnight. After washing twice with PBS (100µL/well), the nonspecific binding sites were blocked by 0.02% gelatin in PBS (60 min, 100µL/well), and the samples were then treated with 1% Tween^®^ 20 (20 min, 100 µL/well, Cat. No. P1379, Sigma-Aldrich, St. Louis, MO, USA). Before the primary antibodies were used, the wells were washed in PBS, then washed twice in PBS with 0.1% Triton X-100 in PBS and were then washed in PBS again. The primary antibodies, namely monoclonal anti-human vinculin (clone hVIN-1, mouse ascites fluid, Cat. No. V9131, Sigma-Aldrich, St. Louis, MO, USA), monoclonal anti-synthetic N-terminal peptide of β-actin (clone AC-15, mouse ascites fluid, Cat. No. A5441, Sigma-Aldrich, St. Louis, MO, USA) and polyclonal rabbit anti-human osteocalcin (peptide 1–49) IgG (Peninsula Laboratories, Bachem Group, San Carlos, CA, USA; Cat. No. T-4743.0400) were diluted to a concentration of 1:200 in PBS and were applied at room temperature (60 min, 50µL/well). After subsequent washing in 0.05% Tween in PBS (3 times), secondary antibodies conjugated with peroxidase were applied for 45 min (100µL/well). Goat anti-mouse F(ab´)2 IgG fragment (Sigma-Aldrich, St. Louis, MO, USA, Cat. No. A3682, dilution 1:1000) was used after the mouse monoclonal primary antibodies, and goat anti-rabbit IgG (Sigma-Aldrich, St. Louis, MO, USA, Cat. No. A9169, dilution 1:5000) was used after the rabbit polyclonal antibody. This step was followed by washing in 0.05% Tween in PBS (3 times) again, and then by an orthophenylendiamine (Sigma-Aldrich, St. Louis, MO, USA, Cat. No. P1526, concentration 2.76 mM) reaction, using 0.05% H_2_O_2_ in 0.1 M phosphate buffer (pH 6.0, dark place, 100 µL/well). The reaction was stopped after 20 min by 2 M H_2_SO_4_ (50 µL/well), and the absorbance was measured at 490 and 690 nm by a Versa Max Microplate Reader (Molecular Devices Corporation, Sunnyvale, CA, USA). The absorbances obtained from the cells growing on UCS and MCS were expressed as a percentage of the values obtained in the control cultures on standard polystyrene dishes.

### 2.6. Potential Immune Activation of Cells on the Scaffolds

The potential immune activation of cells in contact with the scaffolds was estimated by the concentration of TNF-α, a pro-inflammatory cytokine, in the cell culture medium after cultivation of murine-macrophage-like RAW 264.7 cells (American Type Culture Collection, Manassas, VA, USA) on the tested scaffolds by a method developed earlier [44,45]. Both UCS and MCS were fixed in CellCrowns, inserted in polystyrene 24-well cell culture plates (TPP, Trasadingen, Switzerland; internal well diameter 1.5 cm) and seeded with 30,000 cells per well into 1.5 mL of RPMI-1640 medium (Sigma-Aldrich, St. Louis, MO, USA), supplemented with 10% of FBS and 40 µg/mL of gentamicin, as mentioned above. Pure polystyrene wells were used as reference samples. After 24 h, the scaffolds were transferred into fresh wells filled with cell culture medium in order to exclude cells adhering to the underlying well bottom from the evaluation. In the reference wells, the culture medium was also changed. The cells were then cultivated for 6 additional days. For each experimental group, two samples were used, and two measurements were performed in each sample.

After 6 days of cultivation, the cell culture medium was collected, and the concentration of TNF-α was measured by a sandwich ELISA using a mouse TNF-α Quantikine ELISA kit (R&D Systems, Minneapolis, MN, USA), according to the manufacturer’s protocol. As a positive control for TNF-α production, we used RAW 264.7 cells grown in polystyrene culture wells and stimulated with lipopolysaccharide (LPS) from *Escherichia coli* (0111:B4, γ-irradiated, Sigma-Aldrich, St. Louis, MO, USA). LPS was applied for 24 h in concentrations of 0.01 or 1.0 μg/mL on day 6 after seeding. For each concentration, two wells were used, and two measurements were performed in each well. The concentrations of TNF-α were expressed in pg per 100,000 cells.

### 2.7. Statistics

The quantitative data were presented as mean ± S.E.M. (standard error of mean). Multiple comparison procedures were performed by the one-way analysis of variance (ANOVA), Student–Newman–Keuls method, using SigmaStat software (Jandel Corp., San Jose, CA, USA). *P* values equal to or less than 0.05 were considered significant. If two experimental groups were used, Student’s *t*-test was applied.

## 3. Results

### 3.1. Morphology of the Scaffolds

The spatial organization of the scaffold materials from their surface towards their base is illustrated graphically on color-coded confocal microscopy projections (Figure 2a,b). The dry unmineralized scaffolds (UCS) were characterized by the even organization of the void space and the collagen fibrous mesh. However, the mineralized collagen scaffolds (MCS) were much denser, and they were covered with a crater-like texture. Similar differences in the morphology of UCS and MCS (i.e., a loose fibrous mesh in UCS and a denser structure in MCS) were also revealed by scanning electron microscopy (Figure 2c,d).

A 3D visualization of the scaffolds by confocal microscopy better demonstrates the spatial distribution of the fibrous mesh (Figure 3a,b). The dry UCS were assembled into an evenly distributed fibrous mesh. The MCS formed a structure that exhibits a shriveled surface with a majority of small entering pore areas and randomly placed crater-like features, i.e., large entering pore areas.

In the wet scaffolds, the structures in the UCS swelled, but the proportion of features remained similar, and only a few small entering pore areas were found. The same trend was observed in the wet MCS; however, the wet environment seems to have had less influence on the fibrous mesh (Figure 3c,d).

Measurements of the pore sizes in confocal microscopy pictures revealed that the dry UCS contain entering pore areas that are closer to a normal distribution. The distribution clearly shows that UCS have a higher number of larger entering pore areas than MCS (Figure 4, Table 1). In UCS, the percentage of pores with entering pore areas ranging from 10^4^ to 10^5^ μm^2^, i.e., approx. 113 μm to 357 μm in diameter (provided that these areas are almost circular, as indicated by Figure 2 and Figure 3), was almost 70% (Table 1). However, only 32.5% of pores 113 μm to 357 μm in diameter were found in the MCS. The majority of the pores in MCS (approx. 63%) were below 10^4^ µm^2^, which establishes a diameter of less than 113 µm. Pores larger than 1000 μm in diameter (i.e., with entering pore areas ranging from 10^4^ to 10^5^ μm^2^) accounted for almost 11% in UCS but only about 4% in MCS (Table 1).

### 3.2. Mechanical Stability of the Scaffolds

The tension force, i.e., the force needed for stretching to rupture the scaffolds did not differ significantly between UCS and MCS, though this force tended to be on an average higher in UCS than in MCS (Figure 5). As the compression test (UCS: 22.35± 0.90, *n* = 10; MCS: 20.46 ± 1.02, *n* = 10, *p* ≤ 0.05), which was conducted in the orthogonal direction to the tensile test, did not return significantly different mechanical properties compared to the tensile test, the material is considered mechanically isotropic. In addition, both types of scaffolds showed similar mechanical stability, i.e., resistance to mechanical damage, and this resistance tended to be slightly higher in the UCS than in the MCS.

### 3.3. Elemental Composition of the Scaffolds

Elemental analysis performed by ICP-OES showed the presence of calcium, potassium, magnesium, sodium and phosphorus in both UCS and MCS. In UCS, the occurrence of these elements can be explained by a contamination of the collagen sample, prepared from bovine skin, with interstitial fluid, a blood-plasma-related body fluid, which is a source of these elements. These elements are also contained in cells physiologically present in the bovine skin tissue. The Ca^2+^, K^+^, Mg^2+^ and Na^+^ cations and phosphate anions can be strongly bound to the collagen molecule by electrostatic interactions, and it can therefore be difficult to remove them completely by the purification process of collagen. However, the immersion of collagen scaffolds in SBF led to a further significant increase in the content of calcium, magnesium and phosphorus and to a decrease in sodium in these scaffolds (Figure 6).

### 3.4. Structural Changes in the Scaffolds after Mineralization

The structural changes in the scaffolds after immersion in SBF were evaluated by FTIR spectroscopy. The infrared spectra of both UCS and MCS are introduced in Figure 7. According to earlier studies performed on collagen mineralized in SBF [9,23], typical collagen peaks at approx. 3300 cm^−1^ and 1600 cm^−1^ are visible in both UCS and MCS; the peak at 3300 cm^−1^ slightly shifted to a lower wavenumber in both types of scaffolds. A relatively weak band at approx. 2900 cm^−1^ in both UCS and MCS represents C-H stretching, which is typical for carbon-based materials including collagen [23]. The band at 1630 cm^−1^ corresponds to the O-H vibration overlapping with the peak of amide I [9].

From Figure 7 it is evident that the differences in the acquired spectra of unmineralized and mineralized collagen are not very significant, but the most noticeable changes (marked with asterisks) are a slight shift of the thick band from 1543.7 cm^−1^ to 1538.2 cm^−1^ and a relative reduction in the band intensity at 1398 cm^−1^ and 1031 cm^−1^ but a slight increase in the intensity at 1080 cm^−1^. The peak at 1540 cm^−1^, as well as the peak at approx. 1200 cm^−1^, visible in both UCS and MCS, arose from the N-H deformation of amide II and C-N stretching of amide III, respectively [9]. The position at 1080 cm^−1^ represents orthophosphate ([PO_4_]^3−^) formation in the scaffold [23]. In other studies, the 1082 cm^−1^ band was assigned to the mutual stretching vibration of the C-O bonds in the carbohydrate as well as to the symmetric PO_2_ stretching vibration in the phosphorylated proteins (for a review, see [46]). The presence of phosphates was also attributed to the bands at 1200–965 cm^−1^ [9], 1022 cm^−1^ [47], 1020 cm^−1^ and 960 cm^−1^ [48] and also at 500–600 cm^−1^ [9,23,48], while the bands at 872 cm^−1^, 873 cm^−1^ and 1413 cm^−1^ were assigned to carbonates [47,48]. Similarly, in KBr-FTIR spectra of a human cortical bone, the bands at 1032, 603 and 563 cm^−1^ were assigned to phosphates, while the band at 872 cm^−1^ was assigned to carbonates [49].

### 3.5. Adhesion, Growth and Maturation of Cells on the Scaffolds

Mineralization of the scaffolds had a positive influence on the adhesion and growth of human-osteoblast-like MG-63 cells. As was revealed by confocal microscopy, the cell population density on the material surface was visibly higher on the MCS than on the UCS (Figure 8), and the cells on MCS also penetrated inside the materials more deeply and in greater numbers (Figure 9). These findings were confirmed by the significantly higher amounts of DNA found in MCS than in UCS on day 1, 3 and 7 after cell seeding (Figure 10).

As revealed by ELISA, the concentration of vinculin, a protein of focal adhesion plaques associated with integrin adhesion receptors, was similar in the cells on UCS, MCS and in the control polystyrene wells (Figure 11a). The concentration of β-actin, a cytoskeletal protein, was also similar in the cells on both types of scaffolds. However, at the same time, the concentration of β-actin in the cells on both types of scaffolds was higher than in the cells on the control polystyrene wells (Figure 11b). The concentration of osteocalcin, a marker of osteogenic cell differentiation/phenotypic maturation, was similar in the cells on both types of scaffolds, but these values were lower than those obtained in the control polystyrene wells (Figure 11c).

### 3.6. Cell Growth in Extracts from the Scaffolds

Real-time monitoring of the cell growth in extracts from the scaffolds using the biosensoric xCELLigence system for 7 days revealed that, with the exception of the first 24 h (Figure 12a), the cell index, an indicator of the coverage of the well bottoms with cells and thus an indicator of cell growth, was higher for cells growing in extracts from MCS than for cells in extracts from UCS (Figure 12b). In addition, the cell index in the extracts from MCS increased continuously, while in UCS the index stagnated or even decreased. On day 7 after seeding, the cells growing in the extracts from MCS were almost confluent, while the cells in the extracts from UCS were very sparse (Figure 13). At the same time, however, the cell index of the extract from MCS was lower than the value obtained in the standard cell culture medium, and on day 7 the cells in the MCS extracts were less confluent (Figure 12b and Figure 13).

### 3.7. Potential Immune Activation of Cells on the Scaffolds

The potential immunogenicity of the scaffolds was evaluated by the production of TNF-α by mouse macrophage-like RAW 264.7 cells in cultures in the presence of scaffolds. The concentration of TNF-α in the culture medium was measured using a commercially available kit and was calculated per 100,000 cells in each tested experimental group. The UCS evoked significantly higher production of TNF-α in the cells than the standard cell culture PS dishes, but this production was significantly lower than in the positive control, i.e., RAW 264.7 cells cultured on PS dishes stimulated by 0.01 and 1.0 μg of bacterial lipopolysaccharide (LPS, Figure 14a). However, the production of TNF-α was markedly higher in the cells on MCS than in the cells on UCS (about 4.5 times higher) and in the cells stimulated with both concentrations of LPS (about 3 times). From this point of view, MCS seem to be significantly more immunogenic than their non-mineralized counterparts.

The number of RAW 264.7 cells on the tested materials was inversely correlated with the production of TNF-α. The lowest number of cells (550 000 ± 33 000 cells/mL) was obtained for MCS, a higher number of cells (985 000 ± 22 000 cells/mL) was obtained for UCS, and the highest number (1 784 000 ± 33 000 cells/mL) was obtained for PS dishes (Figure 14b). At the same time, the number of RAW 264.7 cells cultured on PS dishes in the presence of LPS was similar to the values on pure PS dishes or was slightly increased where there was a higher concentration of LPS (Figure 14b). Thus, it is apparent that LPS did not greatly enhance the proliferation of RAW 264.7 cells.

## 4. Discussion

### 4.1. Properties of the Scaffolds and Their Colonization with Cells

In this study, porous sponge-like scaffolds, intended for bone tissue engineering, were prepared from bovine collagen I by freeze-drying and were either left unmineralized (labelled as UCS) or were mineralized in a simulated body fluid (labelled as MCS). As revealed by confocal microscopy and SEM (Figure 2 and Figure 3), the UCS were looser and contained a higher percentage (almost 70%) of larger pores 113 μm to 357 μm in diameter. In contrast, MCS were denser and contained a higher percentage (approx. 63%) of pores smaller than 113 µm in diameter (Figure 4, Table 1). An explanation for this could be the obstruction of pores in MCS with an increased content of Ca, Mg and P (Figure 6), which can form calcium and magnesium phosphates, e.g., hydroxyapatite (HAp) and apatite. Similar results were published in a study by Al-Munajjed et al. [23], where unmineralized porous collagen scaffolds showed a loose structure with large, pronounced pores, while the scaffolds immersed in SBF were more compact and contained HAp. In another study by Xia et al. [9], where mineralized collagen scaffolds were generated by collagen self-assembly and in situ apatite precipitation in a collagen-containing modified SBF, the compactness of the scaffolds increased (and the size of pores within these scaffolds decreased) with an increasing concentration of apatite.

Another important cause of a denser structure of MCS in comparison with UCS could be a partial collapse and shrinkage of the inner architecture of the collagen foam due to a hydrolytic degradation of collagen in the Tris–HCl buffer during mineralization. Surprisingly, a fast hydrolytic degradation of three different commercial porcine collagen matrices, based on native dermal type I and III collagen, was recently described by Vallecillo et al. [50]. After 48 h of immersion of the samples in the phosphate-buffered saline (PBS) at 37 °C, the authors observed considerable volume and weight losses of collagen matrices—all the three matrices experienced a loss of weight between 40 and 80%. The hydrolytic degradation of collagen in the Tris–HCl buffer during mineralization could also explain, at least partly, why the mechanical stability of MCS in our study was not improved but tended to be even slightly worse than in UCS.

For the formation of regenerated bone tissue within the scaffolds, pores larger than 100 μm in diameter are required. The reason is that for proper bone formation, osteogenic cells need a similar environment as they would experience in natural bone tissue. In cortical (compact) bone, which forms the hard outer layer of bones, the diameter of osteons (i.e., Haversian systems of concentric layers of osteoblasts and osteocytes around a central canal with blood supply) is minimally 100–200 µm. In cancellous (trabecular or spongy) bone, which is the internal tissue of bones, the size of the pores is even larger, up to 600 µm (for a review, see [7,9,51]). A relatively large size of pores is required for proper accommodation of osteoblasts within the scaffolds, i.e., for their adhesion, spreading, migration and proliferation as well as the production of mineralized bone matrix and formation of osteon-like structures. However, the majority of the pores in MCS (approx. 63%) were less than 113 µm in diameter (Figure 4, Table 1). Pores 75–100 μm in diameter have been reported to promote the ingrowth of unmineralized osteoid tissue, and pores smaller than 75 μm in diameter were filled only by fibrous tissue (for a review, see [7]). In our earlier studies performed on MG-63 cells seeded on porous poly(lactide-*co*-glycolide) (PLGA) scaffolds, pore entrances 40 μm in diameter were often spanned by these cells, which prevented the colonization of the scaffold interior with cells. The optimum pore size for the ingrowth of MG-63 cells inside the PLGA scaffolds was within the range from 400 to 600 μm [52]. In accordance with this, some other researchers have suggested that the optimum pore size for the ingrowth of osteoblasts is in the range from 300 to 500 μm (for a review, see [9]).

In spite of the smaller pore size in the MCS, the colonization of these scaffolds with MG-63 cells on their surface as well as inside was significantly higher than that of their unmineralized counterparts (Figure 8, Figure 9 and Figure 10). This result seems to be rather surprising, because a smaller pore size has often been associated with lower cell colonization of various types of scaffolds [52]; for a review, see [9]). However, in a study by Xia et al. [9], composite collagen–HAp scaffolds remarkably accelerated new bone formation in calvarial defects in mice in comparison with pure collagen scaffolds, although the pore size in the composite scaffolds was smaller and decreased with increasing HAp content. One explanation was that small pores may result in a large surface area available for cell adhesion. It is known that at a constant porosity, i.e., the percentage of void space in scaffolds, the scaffolds with smaller pore diameters possess a larger inner-surface area that can be used for cell colonization (provided that the pore size is sufficient for the cell ingrowth) [8,9]. Similarly, on the outer surface of these scaffolds, cells can also find more sites for their attachment and can bridge the void spaces more easily than on surfaces with a large pore diameter [8,9]. As was revealed by confocal microscopy and SEM, the surface of the MCS in our study seemed to be more organized and more compact (Figure 2, Figure 3 and Figure 4) and it could therefore provide more stable and more continuous growth support for cells during their spreading and migration than in the case of UCS. In addition, the MCS contained a lower percentage of pores more than 1000 μm in diameter (4% versus 11% in UCS; Figure 4, Table 1). These big pores may not provide sufficient surface area for cell attachment to the scaffolds and are usually not entirely filled with regenerated mineralized bone tissue [6,9].

In a study by Antebi et al. [29], mineralized collagen scaffolds also provided better growth support for cells than non-mineralized collagen scaffolds. The mineralized scaffolds were homogeneously infiltrated with human mesenchymal stem cells, while in the non-mineralized scaffolds, the cells were only seen on the periphery of the construct, with very little cellularity in the center of the construct. This result was explained by the higher mechanical stability of the mineralized scaffolds, which prevented the shrinkage and collapse of the scaffolds by cell-mediated contraction [29]. However, in our study, the mechanical stability, as evaluated by the tensile and compression tests, was similar in both MCS and UCS (Figure 5). This is in accordance with our finding of a similar concentration of vinculin in the cells growing on both types of scaffolds (Figure 11a). Vinculin is a focal adhesion protein stabilizing the focal adhesion plaques, so it can be considered as a marker of good adhesion strength of cells. In our earlier study, talin- and vinculin-containing focal adhesion plaques were better developed in cells growing on nanofibrous PLGA membranes reinforced with diamond nanoparticles than on pure PLGA membranes. This was attributed to the superior mechanical properties of the PLGA–nanodiamond composite membranes [44].

The concentration of β-actin, a cytoskeletal protein, was also similar in the cells on both types of scaffolds but, in these cells, the concentration was higher than in the cells on the control polystyrene dishes (Figure 11b). Similar results were obtained in our earlier study performed on MG-63 cells on pure polysulfone foils and on foils containing various concentrations of carbon nanohorns or carbon nanotubes. The cells on all of these substrates contained well-developed β-actin cables, reminiscent of those present in cells of muscle type and more apparent than in cells on harder and stiffer substrates, e.g., microscopic glass coverslips. These findings were explained by the relatively low stiffness of polysulfone, even after it has been reinforced with carbon nanoparticles [53]. A similar explanation could also be used in our present study. The stiffness of the cell adhesion substrate is a decisive factor directing the cell differentiation towards a certain phenotype. For example, on very soft polyacrylamide gels, mimicking the mechanical properties of soft brain tissue, human mesenchymal stem cells (MSC) differentiated towards the neuronal phenotype. On harder gels, mimicking the mechanical properties of muscle tissue, the MSCs acquired a myogenic phenotype, and only very stiff matrices were osteogenic [54]. In accordance with these findings, the concentration of osteocalcin, a marker of osteogenic cell differentiation, was similar in cells on both non-mineralized and mineralized collagen scaffolds (Figure 11c), i.e., on materials with similar mechanical properties, and was lower than in cells on polystyrene dishes, which were stiffer than the deformable collagen-based scaffolds.

### 4.2. Positive Effect of Scaffold Mineralization on Cell Colonization

Since the mechanical properties of UCS and MCS are similar and the pore size, which facilitates the ingrowth of cells inside the scaffolds, is even lower in MCS, the stimulatory effects of MCS on the growth of MG-63 cells in our study could be attributed to a direct biochemical action of the mineral component of the scaffolds. Elemental analysis showed a significant increase in the content of calcium, magnesium and phosphorus in the MCS (Figure 6). It is well known that calcium is the simplest and most versatile second messenger mediating processes associated with the adhesion, migration, proliferation, survival, apoptosis, differentiation and phenotypic maturation of cells (for a review, see [55,56,57,58]). Ca^2+^ ions are essential for adhesion function of integrins and other cell–matrix adhesion receptors for the formation of focal adhesion plaques and actin cytoskeleton, structures necessary for cell attachment, spreading and migration [55,56]. Calcium is needed for the expression of genes involved in cell growth, for the activation of tyrosine kinase receptors and for the cell cycle progression [57,58]. Magnesium and phosphorus also play positive roles in cell adhesion, migration and proliferation, key processes for cell ingrowth inside the scaffolds. Magnesium can act synergistically with calcium in its activities supporting cell adhesion and proliferation [59]. Similarly, phosphorus also acts as an important activator of cell proliferation. Elevated extracellular inorganic phosphate resulted in the induction of proliferation-associated genes, such as c-fos [60]. It is also generally known that the phosphorylation of specific substrates by kinase enzymes, e.g., mitogen-activated protein kinases or cyclin-dependent kinases, is a key factor for the induction of cell proliferation. Taken together, Ca, Mg and P and their compounds have a considerable osteoconductive effect, which lies in the stimulation of adhesion, migration and proliferation of osteoblasts, and also an osteoinductive effect, i.e., the ability to induce differentiation of immature cells, e.g., stem cells, towards osteoblasts [61,62].

In contrast to calcium, magnesium and phosphorus, the content of sodium was decreased in MCS compared to UCS (Figure 6). This decrease could be explained by the release of Na from the scaffolds. In calcium phosphates, e.g., HAp, calcium ions can be, at least partly, substituted by sodium ions, but this substitution increases the instability, solubility and degradability of calcium phosphates, followed by the release of sodium [63,64]. Substitution of calcium with alkali metals (sodium, potassium, strontium) has been used for increasing the osteoconductivity of calcium phosphates, i.e., materials used for bone tissue engineering and regeneration. For example, potassium–strontium or sodium–strontium co-substituted calcium polyphosphate bioceramics increased the proliferation activity of rat-osteoblast-like cells in vitro in comparison with non-substituted ceramics [63]. Similarly, sodium-substituted HAp exhibited markedly higher osteoconductivity than non-substituted HAp when implanted in calvarial defects of New Zealand white rabbits in vivo [64]. These positive effects were attributed to a higher dissolution and degradation ability of the Na-substituted phosphates. In accordance with this, our MCS also improved cell growth directly on the scaffolds as well as in extracts from these scaffolds.

In contrast to elemental analysis, FTIR spectroscopy did not reveal significant differences between UCS and MCS (Figure 7). The bands typical for phosphates and carbonates were not considerably changed by the mineralization of collagen in our present study. It can be attributed to the fact that our mineralization process lasted for a relatively short period, i.e., only 48 h. Even in collagen gels where the mineralization in SBF was enhanced by citric acid, the phosphates started to be detectable after only 3 days of mineralization and were clearly apparent after 7 days, and the carbonates were detectable after only 7 days [47]. A clear time-dependent mineralization of collagen was also shown by dynamic attenuated total reflection (ATR)–FTIR spectroscopy in collagen sponges subjected for 7 days to an intensive biomimetic intrafibrillar mineralization using a negatively charged polyelectrolyte [48].

In spite of a relatively low level of mineralization, which was under the detection level of FTIR spectroscopy, the mineral component was able to stimulate the proliferation of MG-63 cells cultured not only directly on the scaffolds but also in extracts from the scaffolds. Real-time monitoring of the cell growth using the biosensoric xCELLigence system revealed a higher and continuously increasing cell index in cells growing in extracts from MCS and their higher final cell population density on day 7 (Figure 12 and Figure 13). However, at the same time, the cell index in the extract from MCS was lower than the value obtained in the standard cell culture medium, and on day 7 the cells in the MCS extracts were less confluent (Figure 12 and Figure 13). These results could be explained by a certain depletion of Ca, Mg and phosphate ions from the medium while the UCS and MCS were being extracted. Ca and phosphate ions can be captured by the carboxyl, hydroxyl and amine functional groups present in collagen molecules, which serve as nucleation sites for the formation of HAp [11]. Depletion of calcium ions from the culture medium has also been reported after it has been exposed to HAp and other calcium phosphates, which resulted in the reduced growth of human osteoblasts [65] or osteoblast-like Saos-2 or MG-63 cells [66]. The ion depletion in the medium was probably less pronounced in the extracts from MCS than in the extracts from UCS, because it could be partly compensated by the release of ions from the MCS.

Interestingly, during the first 24 h after seeding, the cell index was higher in the extracts from UCS than in the extracts from MCS (Figure 12a). A similar result was obtained in our earlier study performed on nanofibrous poly(L-lactide) (PLLA) membranes loaded with 15 wt.% of HAp nanoparticles. The colonization of these membranes with MG-63 cells was delayed in the first 3 days of cultivation in comparison with the pure membranes, although on day 7 the cells on the HAp-loaded membranes reached a significantly higher population density [45]. It should also be taken into account that the cell index is proportional not only to the cell number but also to the cell spreading, i.e., to the cell–material contact area. Cell attachment and cell spreading are predominant processes during the first day after cell seeding, when the cells are in the so-called lag phase and usually do not proliferate. The process of initial cell attachment and spreading might differ in the UCS and MCS extracts, which is a topic that needs to be further investigated. In a study by Obata et al. [59], performed on MC3T3-E1 mouse pre-osteoblasts, the number of initially attached cells, their spreading and their subsequent growth were either enhanced or reduced depending on the concentration and various combinations of Ca, Mg and Si ions added into the cell culture medium [59]. A lower concentration of Na in MCS might also contribute to lower cell adhesion in extracts from these scaffolds via changes in the expression of α- and β-subunits of sodium channels, which can act as molecules regulating cell adhesion. Specifically, the β-subunits acted as cell adhesion-mediating molecules enhancing the cell adhesion through interaction with extracellular matrix molecules and the cell cytoskeleton, while the α-subunits were associated with lower adhesiveness, higher migration potential and greater invasiveness of cancer cells [67,68].

### 4.3. Extrafibrillar versus Intrafibrillar Mineralization of Collagen

The positive effect of scaffold mineralization on cell growth observed in our study could be further enhanced by intrafibrillar mineralization of the scaffold. Simple immersion of the collagen scaffold into SBF, as carried out in our study, is known to lead mainly to extrafibrillar mineralization of the collagen matrix, i.e., deposition of the mineral component on the surface of collagen fibrils [10,29,30]. In order to achieve intrafibrillar mineralization, i.e., deposition of minerals within the gap zone of collagen fibrils, which occurs in the native bone tissue, it is necessary to simulate the natural process of collagen mineralization. In the natural process, the mineral component is interconnected with collagen molecules through non-collagenous proteins, i.e., osteocalcin, osteopontin and bone sialoprotein, which are generally highly negatively charged due to the abundance of acidic amino acids in their molecules [24]; for a review, see [10,11]). This phenomenon inspired the development of the polymer-induced liquid precursor (PILP) process, where the mineralization of collagen in ionic solutions is facilitated by the presence of various anionic polymers, such as poly-L-aspartic acid, poly-L-glutamic acid, polyvinylphosphonic acid and polyacrylic acid ([24]; for a review, see [10,11]). Additionally, other negatively charged compounds, such as citric acid [47], sericin [69], carboxymethyl chitosan [30], and also fluid shear stress, generated, e.g., by dynamic cell culture systems [11], have been shown to have a positive effect on intrafibrillar collagen mineralization.

Intrafibrillary deposited mineral components obstruct the pores within collagen scaffolds to a much lesser extent than the minerals deposited mainly extrafibrillary or even admixed artificially into the scaffolds [29], and thus intrafibrillary mineralized scaffolds can have a more appropriate pore size for the immigration and growth of osteoblasts. Intrafibrillar mineralization can also improve the mechanical properties of the scaffolds in comparison with extrafibrillar mineralization [30,48] or in comparison with the incorporation of mineral particles [29]. For example, collagen scaffolds mineralized by PILP using carboxymethyl chitosan as a negatively charged polymer showed a higher modulus of elasticity than scaffolds mineralized traditionally by immersion in SBF [30]. The improved mechanical properties then had supportive effects on cell adhesion and spreading, on the assembly of the actin cytoskeleton and particularly on osteogenic cell differentiation [33].

### 4.4. Immunological Acceptance of the Scaffolds

Collagen scaffolds for tissue engineering are usually made of collagen of animal origin, e.g., bovine or porcine. Although bovine or porcine collagen is approved for biomedical use, including implantation or injection into patients, it is recognized as a foreign material and can evoke immune reactions in humans [34] and laboratory animals, such as mice [35] or rats [36]. These immune reactions are mediated by the cells of non-specific (innate) immunity, such as leukocytes and macrophages, and by the cells of specific immunity (acquired during the lifetime of an organism), such as B and T lymphocytes. The cells of the non-specific immunity eliminate the foreign material by phagocytosis and by digestion in lysosomes, while the cells of the specific immunity produce antibodies (B-lymphocytes) or various enzymes, including proteases (T-lymphocytes). The cells of both non-specific and specific immune systems produce cytokines, i.e., signaling molecules, which can be pro-inflammatory (i.e., attracting additional immune cells and stimulating their proliferation, maturation and function) or anti-inflammatory (i.e., further modulating and reducing the inflammatory response). A typical example of pro-inflammatory cytokines is tumor necrosis factor-alpha (TNF-α), the production of which is an important marker of cell immune activation (for a review, see [70]).

Intrafibrillar mineralization of collagen scaffolds, as described above, can improve the immunological acceptance of these scaffolds. It can alleviate the inflammatory reaction to the foreign materials, particularly that mediated by macrophages. In a study by Sun et al. [39], intrafibrillary mineralized collagen scaffolds implanted into mandible defects of rats in vivo attracted more anti-inflammatory M2 macrophages, characterized by the expression of CD68 and CD163 and associated with the resolution phase of inflammation and with the repair of damaged tissues. On the contrary, extrafibrillary mineralized collagen scaffolds attracted predominantly pro-inflammatory M1 macrophages, expressing CD68 and inducible nitric oxide synthase [39]. Similar results were also obtained in an in vitro study performed on mouse-macrophage-like RAW 264.7 cells cultured on collagen mineralized in solutions of CaCl_2_ and H_3_PO_4_ and on pure hydroxyapatite. On the mineralized collagen, the RAW 264.7 cells differentiated towards anti-inflammatory M2 macrophages, as indicated by positive staining for CD206 and by the production of IL-10, while on hydroxyapatite, the RAW 264.7 cells differentiated towards pro-inflammatory M1 macrophages, as indicated by the expression of CD86 and the secretion of TNF-α by these cells [40].

In our present study, the production of TNF-α, a pro-inflammatory cytokine, was significantly higher in the RAW 264.7 cells cultured on MCS than in the cells on UCS (Figure 14a), which suggests predominantly extrafibrillar mineralization of collagen in our scaffolds. It is known that calcium phosphates used in direct contact with cells and tissues, e.g., as ceramic implants or as biomaterial coatings, evoke an inflammatory response, manifested by the recruitment of leukocytes, monocytes and macrophages, the production of cytokines, chemokines and metalloproteases and the differentiation and activity of osteoclasts. These reactions are triggered by calcium phosphate particles, which are generated by a dissolution/precipitation process that occurs at the interface between the cells and calcium phosphate materials (for a review, see [71]). A moderate biomaterial-mediated immune response is important for the recruitment of stem cells, tissue regeneration and integration of the implant [72], but a more pronounced inflammatory cell response can lead to loosening of the implant [71]. From this point of view, it seems reasonable to isolate calcium phosphates, at least partly, to avoid direct contact with cells, e.g., by embedding these minerals into collagen by intrafibrillar mineralization. In our earlier study performed on poly(L-lactide) nanofibers loaded with HAp nanoparticles, RAW 264.7 cells produced similar quantities of TNF-α as on pure PLLA nanofibers, and these quantities were significantly lower than in RAW 264.7 cells stimulated with bacterial LPS. This result could be attributed to the fact that a considerable number of HAp nanoparticles were embedded inside the polymer nanofibers [45].

Interestingly, the number of RAW 264.7 cells on the tested materials was inversely correlated with the production of TNF-α, i.e., it was lowest on MCS, medium on UCS and highest on the control PS wells (*cf.* Figure 14a,b). Similar results were obtained in a study by Matesanz et al. [73], where RAW-264.7 cells and mouse peritoneal macrophages experienced significantly decreased proliferation and phagocytic activity when cultured in the presence of nanocrystalline HAp or Si-substituted HAp [73]. These findings were explained by a significant decrease in Ca^2+^ in the culture medium produced by both nano-HAp and nano-SiHAp. On biphasic calcium phosphate granules, the decrease in the number of macrophages and also osteoblasts was attributed to cell membrane lysis, due to dissolution of the material surface, resulting in high phosphate levels and low calcium levels in the immediate cell culture environment [74]. However, these phenomena are less probable in the cells on our scaffolds, because they only occurred in the RAW 264.7 macrophages and not in the MG-63 osteoblasts, which were markedly more numerous on the mineralized scaffolds. Another possible explanation is, therefore, the autocrine regulation of the number of RAW 264.7 by the secretion of TNF-α. It is known that TNF-α itself does not significantly stimulate the proliferation of macrophages, or even inhibits it. TNF-α stimulated the proliferation of macrophages only in synergy with other cytokines, particularly the macrophage colony-stimulating factor [75]. Autocrine production of TNF-α in macrophages reduced their proliferation and induced their terminal differentiation [76]. TNF-α also inhibited the proliferation of mouse myeloid leukemic WEHI-3B JCS cells [77] and human promyelocytic HL-60 leukemia cells [78], inducing their differentiation into monocytes and macrophages. It should be pointed out that RAW 264.7 cells are also of leukemic origin, i.e., they were established from an ascites of a tumor induced in a mouse by an intraperitoneal injection of Abelson leukemia virus [79].

The number of RAW 264.7 cells cultured on PS dishes in the presence of LPS was similar to the values on pure PS dishes or was slightly increased where there was a higher concentration of LPS (Figure 14b). It is therefore apparent that LPS did not greatly enhance the proliferation of RAW 264.7 cells. In studies by other authors, LPS acted rather as a weak or indirect stimulator of the growth of macrophages and related cell types. For example, exposure of murine peritoneal macrophages to LPS (20 ng/mL) induced a transient expression of c-fos and c-myc oncogenes but did not induce DNA synthesis [80]. LPS from *E. coli* greatly stimulated phagocytosis in mouse-bone-marrow-derived macrophages but it did not stimulate the proliferation of these cells [81]. Stimulation of murine bone-marrow-derived macrophages with LPS arrested the growth of these cells and engaged them in a pro-inflammatory response [82]. In macrophage-like RAW 264.7 cells, LPS stimulated differentiation towards osteoclast phenotype in synergy with TNF-α [83]. Thus, LPS induces immune activation, differentiation and functioning rather than proliferation of macrophages.

## 5. Conclusions

Porous sponge-like collagen scaffolds were prepared by freeze-drying and were biomimetically mineralized in simulated body fluid. The mineralization of the scaffolds was verified by elemental analysis, which revealed an increase in the content of Ca, Mg and P, and a decrease in Na, although FTIR spectroscopy did not reveal any significant structural changes in the scaffolds. The mechanical properties, evaluated by tensile and compression tests, were similar in the pure scaffold and in the mineralized scaffolds. However, the growth of human-osteoblast-like MG-63 cells, as estimated by the DNA content in three time intervals, was higher on the mineralized scaffolds. The cells on the mineralized scaffolds also penetrated deeper and in greater numbers inside the scaffolds, although the mineralized scaffolds contained more pores of smaller diameter. From day 1 to 7, the cells grew more rapidly in extracts from the mineralized scaffolds. However, mouse macrophage-like RAW 264.7 cells produced more TNF-α on the mineralized scaffolds, and this production was even higher than in the cells on polystyrene dishes after stimulation with bacterial lipopolysaccharide in concentrations of 0.01 μg/mL or 1 μg/mL. Thus, mineralized collagen scaffolds are more suitable growth supports for bone cells than unmineralized collagen scaffolds, but their potentially high immunogenicity should be taken into account in bone tissue engineering. However, this immunogenicity could be considerably reduced if intrafibrillar mineralization of collagen scaffolds is achieved.

## Figures and Tables

**Figure 1 polymers-14-00602-f001:**
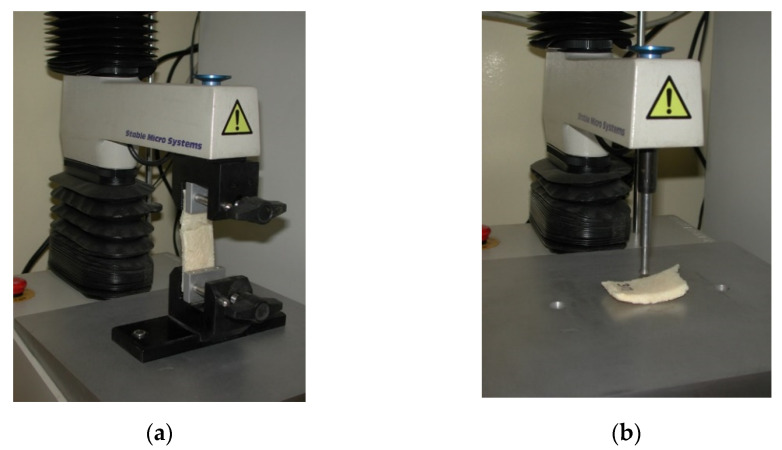
TA-XT2i texture analyzer measuring force at tension (**a**) and compression (**b**).

**Figure 2 polymers-14-00602-f002:**
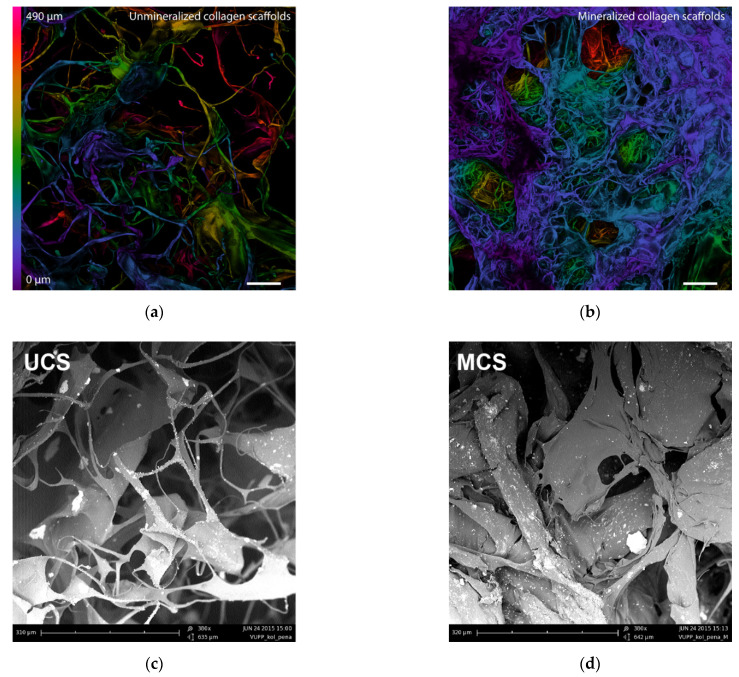
Color-coded confocal microscopy images (**a**,**b**) and scanning electron microscopy images (**c**,**d**) of dry UCS (**a**,**c**) and MCS (**b**,**d**) scaffolds. (**a**,**b**) The depth of various structures within the scaffolds is indicated by spectral colors (see the bar on the left of the image **a**). Summarization of 23 horizontal optical sections recorded every 22 μm. Leica SPE DM2500 confocal microscope, obj.10×. Bar = 100 µm. (**c**,**d**) Phenom™ G2 desktop scanning electron microscope images. Original magnification 380x, bar = 310 µm.

**Figure 3 polymers-14-00602-f003:**
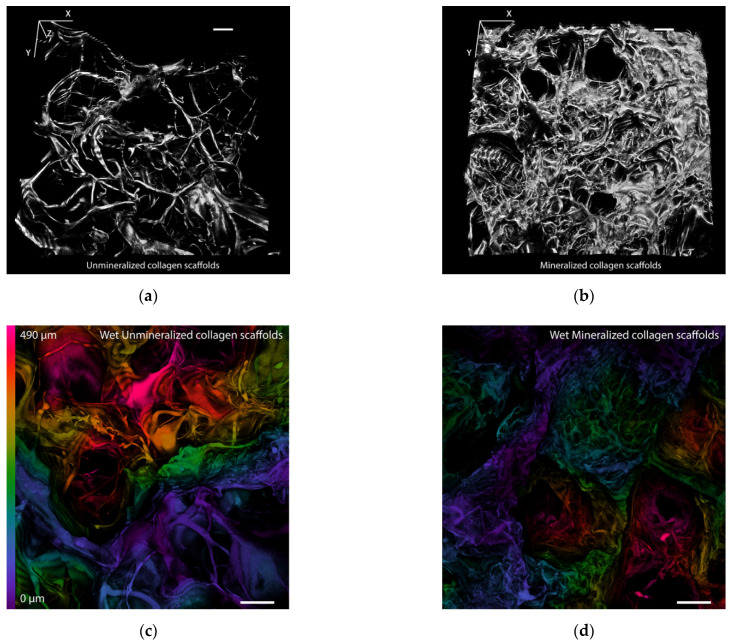
Confocal microscopy images of UCS (**a**,**c**) and MCS (**b**,**d**). Images (**a**,**b**) are 3D visualizations of the dry scaffolds. Images (**c**,**d**) are color-coded images of the wet scaffolds. The depth of various structures within the scaffolds is indicated by spectral colors (see the bar on the left of image c). Summarization of 23 horizontal optical sections recorded every 22 μm. Leica TCS SP2 confocal microscope, obj. 10x. Bar = 100 µm.

**Figure 4 polymers-14-00602-f004:**
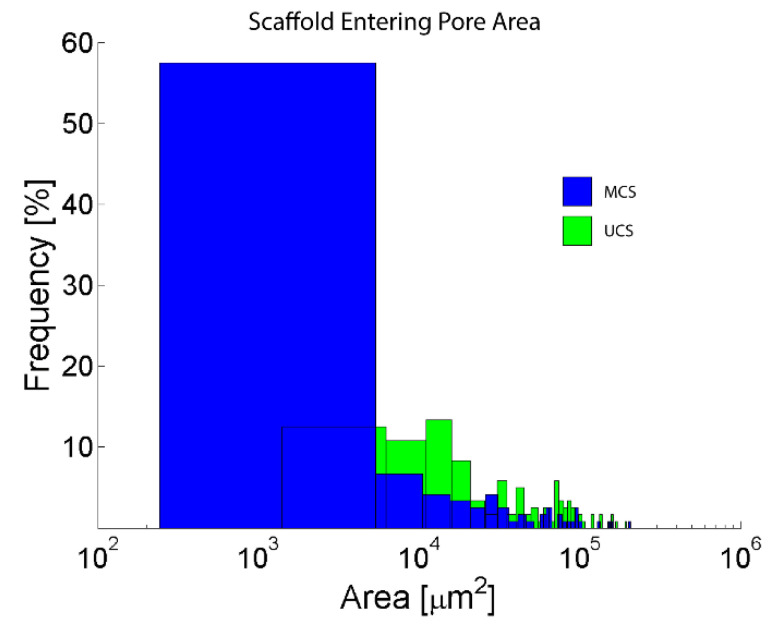
The size and distribution of the entering pore areas in the UCS and MCS. The total number of measured entering pore areas was 120 (15 randomly selected pores in 8 different regions) for each type of scaffold.

**Figure 5 polymers-14-00602-f005:**
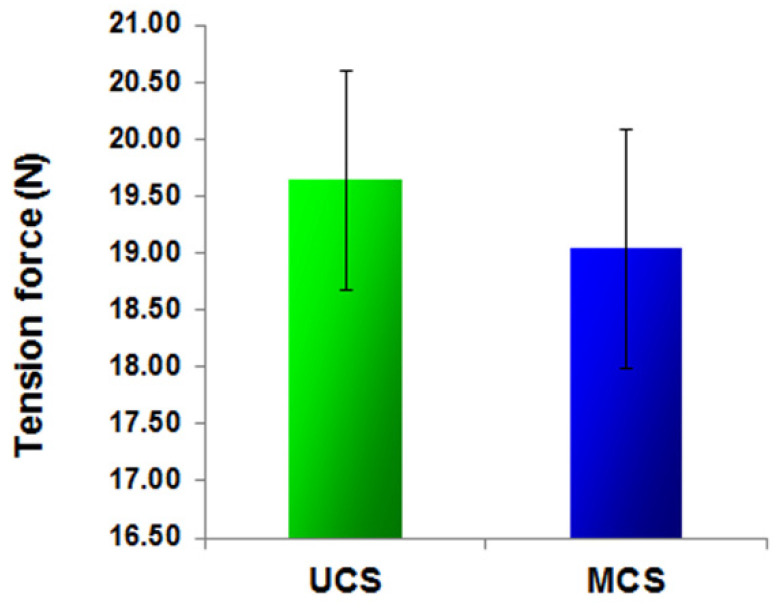
Measurements of tension force, using the TA-XT2i texture analyzer (see Figure 1). Force (newtons) in tension measured at the moment when the sample ruptured on UCS and MCS; mean ± S.E.M. from 3 independent samples. Student’s *t*-test, statistical significance (*p* ≤ 0.05); the test did not reject the null hypothesis.

**Figure 6 polymers-14-00602-f006:**
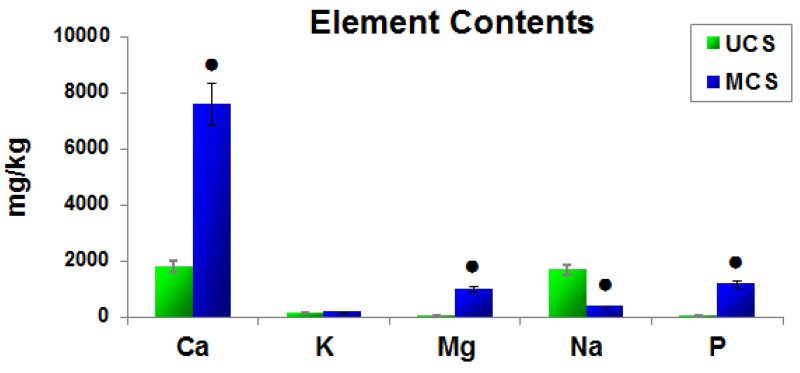
Changes in the element contents of scaffolds before biomimetic mineralization (UCS) and after biomimetic mineralization (MCS). Mean ± S.E.M. from 3 measurements for each element and type of scaffold. Student *t*-test, statistical significance (*p* ≤ 0.05); ● there is a statistically significant difference between the two input groups.

**Figure 7 polymers-14-00602-f007:**
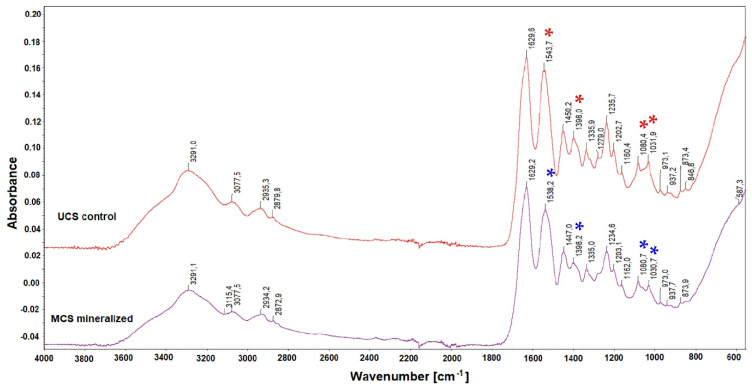
Infrared spectra of unmineralized (UCS) and mineralized (MCS) scaffolds. The most pronounced differences between UCS and MCS spectra are indicated by asterisks.

**Figure 8 polymers-14-00602-f008:**
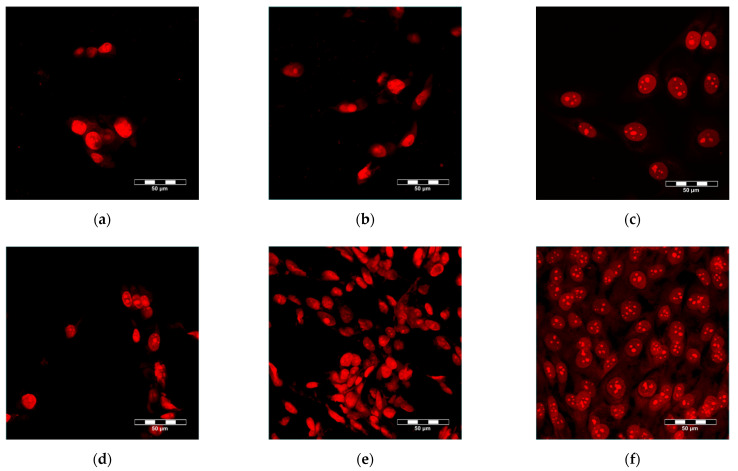
MG-63 cells on day 3 (**a**–**c**) and on day 7 (**d**–**f**) after seeding on UCS (**a**,**d**), MCS (**b**,**e**) and the reference material, i.e., microscopic glass coverslips (**c**,**f**). Cells stained with propidium iodide. Leica TCS SP2 confocal microscope, obj. 20x, zoom 4x, bar = 50 µm.

**Figure 9 polymers-14-00602-f009:**
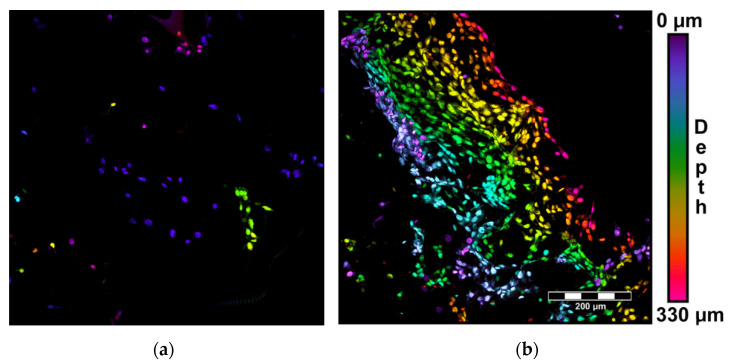
Color-coded projection of cell ingrowth within UCS (**a**) and MCS (**b**) on day 7 after seeding. Cells stained with propidium iodide. The depth of the cells within the scaffolds is indicated by spectral colors (the bar on the right). Summarization of 33 horizontal optical sections recorded every 10 μm. Leica TCS SP2 confocal microscope, obj. 20×, bar = 200 µm.

**Figure 10 polymers-14-00602-f010:**
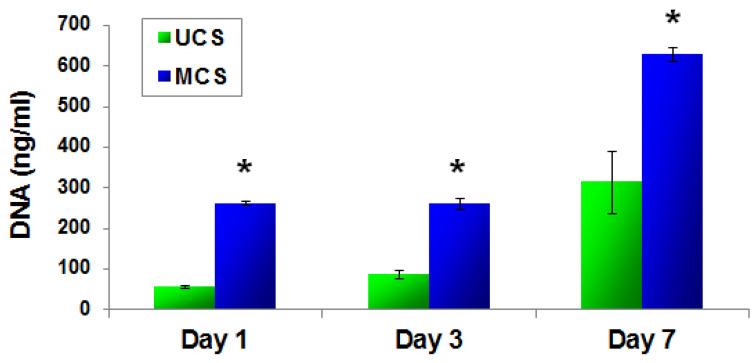
Quantity of dsDNA in MG-63 cells on day 1, 3 and 7 after seeding on UCS and on MCS. Measured using the PicoGreen Assay. Mean ± S.E.M. from 3 independent samples for each experimental group (in each sample, 3 measurements were performed). Student’s *t*-test, statistical significance: * *p* ≤ 0.05 in comparison with UCS.

**Figure 11 polymers-14-00602-f011:**
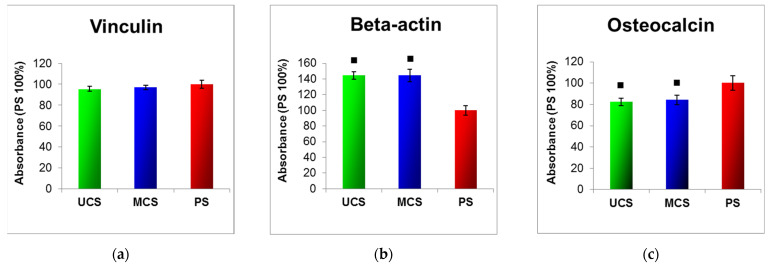
Concentration of vinculin (**a**), β-actin (**b**) and osteocalcin (**c**) in MG-63 cells on day 7 after seeding on UCS, MCS and the control polystyrene (PS), evaluated by an enzyme-linked immunosorbent assay (ELISA) per mg of protein. The absorbance was expressed as a % of the value obtained in cells from the control PS well. Mean ± S.E.M. from 3 independent samples for each experimental group (in each sample, 3 measurements were performed). ANOVA, Student–Newman–Keuls method, statistical significance (*p* ≤ 0.05); **■** Values significantly different from those obtained from PS.

**Figure 12 polymers-14-00602-f012:**
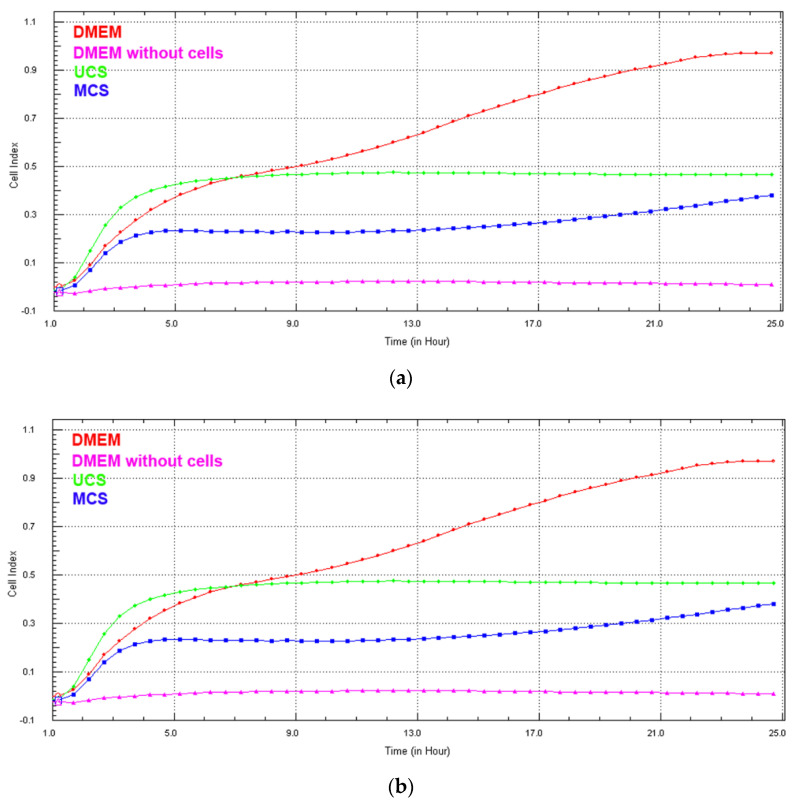
Real-time monitoring of MG-63 cells growing in the standard cell culture medium (DMEM), in an extract from mineralized collagen scaffolds (MCS) and in an extract from unmineralized collagen scaffolds (UCS), during the first 24 h (**a**) and during 7 days (**b**) using the sensory xCELLigence system.

**Figure 13 polymers-14-00602-f013:**
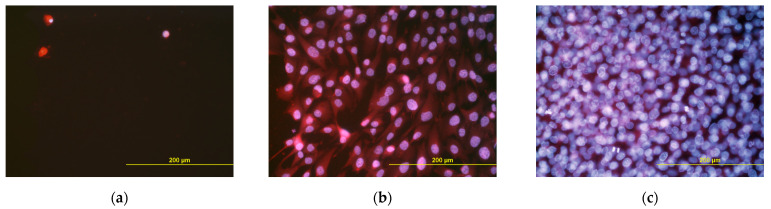
MG-63 cells growing in an extract from unmineralized collagen scaffolds (**a**) in an extract from mineralized collagen scaffolds (**b**) and in the standard cell culture medium (**c**) on day 7 after seeding. Cells stained with Hoechst #33342 and Texas Red C_2_-maleimide fluorescent dyes. Olympus IX 51 microscope, DP 70 digital camera, obj. 20×, bar = 200 µm.

**Figure 14 polymers-14-00602-f014:**
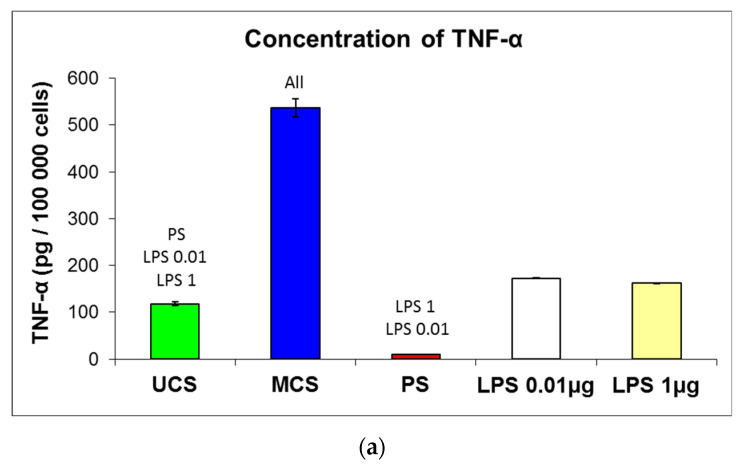
Concentration of TNF-α in the culture medium calculated per 100,000 cells (**a**) and the number of mouse macrophage-like RAW 264.7 cells (**b**) on day 7 after seeding on unmineralized collagen scaffolds (UCS), on mineralized collagen scaffolds (MCS), in polystyrene (PS) wells and in PS wells after stimulation of the cells with bacterial lipopolysaccharide applied on day 6 after seeding for 24 h into the culture medium at a concentration of 0.01μg/mL or 1 μg/mL (LPS 0.01 and LPS 1, respectively). Mean ± S.E.M. from 4 measurements (**a**) or from 100 measurements (**b**) from 2 independent samples for each experimental group. ANOVA, Student–Newman–Keuls method, statistical significance: *p* ≤ 0.05 in comparison with the samples indicated above the columns.

**Table 1 polymers-14-00602-t001:** The percentage of pores of various sizes in the UCS and MCS.

Area [µm^2^]	UCS		MCS	
	Counts	Percent	Counts	Percent
10^2^–10^3^	0	0	11	10
10^3^–10^4^	24	20	64	53.3
10^4^–10^5^	83	69.2	39	32.5
10^5^–10^6^	13	10.8	5	4.2

## Data Availability

The data presented in this study are available on request from the corresponding author.

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
