# Peer review of "Influence of Biomimetically Mineralized Collagen Scaffolds on Bone Cell Proliferation and Immune Activation"

_polymers, 2022, doi:10.3390/polym14030602_

Round 1

Reviewer 1 Report

1. Abstract: Line 22: “However, MSC evoked a more pronounced inflammatory response than UCS…….”. Typing mistake of the word used to indicate mineralized scaffolds. Earlier in the abstract, it is mentioned as MCS. However, in line 22 it is written as MSC.  

2. There are 10 keywords mentioned in the manuscript. Check in the manuscript guidelines about the maximum number of keywords that need to be included in the manuscript. 

3. FT-R Analysis of the UCS and MCS scaffolds need to be included in the revised manuscript. 

4. Authors need to include the reason for the statement mentioned in line 291. “…………loose fibrous mesh in UCS and a more dense structure in MCS”. 

5. Mention the reason for the following statement in line 334. “Both types of scaffolds, therefore, showed similar mechanical stability, which tended to be even higher in the UCS scaffolds than in the MCS scaffolds”. 

6. Why the sodium ion concentration in the UCS was more?

7. Mention the reason why the occurrence of Calcium, Potassium, Magnesium, Phosphorus, and Sodium is found in the unmineralized scaffolds.

8. Mention the reason why cells penetrated deeper in MCS as compared to UCS.

Author Response

The authors thank Reviewer 1 for his (her) valuable comments. All these comments have been carefully considered, and the paper has been modified according to these suggestions (the changes are highlighted in yellow color in the enclosed manuscript).

  1. Abstract: Line 22: “However, MSC evoked a more pronounced inflammatory response than UCS…….”. Typing mistake of the word used to indicate mineralized scaffolds. Earlier in the abstract, it is mentioned as MCS. However, in line 22 it is written as MSC.  

Reply: The referee is right; „MSC“ is a typographical error. The correct abbreviation is MCS, i.e. mineralized collagen scaffolds. MSC is an abbreviation widely used for mesenchymal stem cells. The error has been corrected throughout the whole manuscript.

  1. There are 10 keywords mentioned in the manuscript. Check in the manuscript guidelines about the maximum number of keywords that need to be included in the manuscript. 

Reply: It is written in the official template for the manuscript: „List three to ten pertinent keywords specific to the article yet reasonably common within the subject discipline“. Ten keywords are therefore allowed for an article.

  1. FT-R Analysis of the UCS and MCS scaffolds need to be included in the revised manuscript. 

Reply: The analysis has been performed by Fourier Transform Infrared (FTIR) Spectroscopy and has been included into the manuscript (Materials and Methods - page 5, paragraph 2.2.4.; Results - pages 11-12, new Figure 7; Discussion - page 20). It can be summarized that the differences in the acquired spectra of unmineralized and mineralized collagen scaffolds are only slightly apparent and are not significant. It is probably due to the fact that the mineralization of our scaffolds in SBF lasted for a relatively short period, i.e. 48 hours only. Even in collagen gels where the mineralization in SBF was enhanced by citric acid, the phosphates started to be detectable only after 3 days of mineralization and were clearly apparent only after 7 days, and the carbonates were detectable only after 7 days (Jiang et al. 2020). A clear time-dependent mineralization of collagen was also shown by dynamic attenuated total reflection (ATR)–FTIR spectroscopy in collagen sponges subjected for 7 days to an intensive biomimetic intrafibrillar mineralization using a negatively charged polyelectrolyte (Song et al. 2019).  

Jiang W et al. J Struct Biol 2020, 212(1), 107592. doi: 10.1016/j.jsb.2020.107592.

Song Q et al. Sci Adv 2019, 5(3) eaav9075. doi: 10.1126/sciadv.aav9075.

  1. Authors need to include the reason for the statement mentioned in line 291. “…………loose fibrous mesh in UCS and a more dense structure in MCS”. 

Reply: In both confocal microscopy and scanning electron microscopy images of the scaffolds (Figures 2 and 3), it is clearly visible that the umnineralized scaffolds (UCS) have a loose, rather fibrous structure with larger void spaces, while the mineralized scaffolds (MCS) are denser, more compact, with smaller void spaces (pores). An explanation is the incorporation of minerals into the scaffolds, i.e. increased content of Ca, Mg and P, which usually form calcium and magnesium phosphates, e.g. hydroxyapatite and apatite. Similar pictures were published in a study by Al-Munajjed et al. (2009), where unmineralized porous collagen scaffolds showed a loose structure with large well-apparent pores, while the scaffolds immersed in simulated body fluid were more compact. As revealed by FTIR in that study, the SBF-mineralized scaffolds contained hydroxyapatite. In another study by Xia et al. (2013), where mineralized collagen scaffolds were generated by collagen self-assembly and in situ apatite precipitation in a collagen-containing modified simulated body fluid, the compactness of the scaffolds increased (and the size of pores within these scaffolds decreased) with increasing concentration of apatite.  

Another important cause of a denser structure of MCS in comparison with UCS can be a partial collapse and shrinkage of the inner architecture of the collagen foam due to a hydrolytic degradation of collagen in Tris-HCl buffer during mineralization. Surprisingly, a fast hydrolytic degradation of three different commercial porcine collagen matrices, based on native dermal type I and III collagen, was recently described by Vallecillo et al. (2021). After 48 hours of immersion of the samples in the phosphate-buffered saline (PBS) at 37°C, the authors observed highly significant volume and weight losses of collagen matrices - all the three matrices experienced a loss of weight between 40 and 80%.

These remarks and new references have been added to the Discussion part of the manuscript (page 17, section “4.1. Properties of the scaffolds, and their colonization with cells”).

The problems of collagen mineralization as a possible cause of a denser structure of mineralized scaffolds are further discussed in the section “4.3 Extrafibrillar versus intrafibrillar mineralization of collagen” (Discussion, page 21). Since the minerals deposited within our collagen scaffolds obstruct the pores in these scaffolds, we suppose that these minerals are deposited mainly extrafibrillary. It is known that a simple immersion of the collagen scaffold into SBF, used in our study, leads mainly to extrafibrillar mineralization of the collagen matrix, i.e. deposition of the mineral component on the surface of collagen fibrils. In order to achieve intrafibrillar mineralization, i.e. mineralization less obstructing the pores within the scaffolds, it is necessary to use the Polymer Induced Liquid Precursor (PILP) technique, where the mineralization of collagen in ionic solutions is facilitated by the presence of various anionic polymers. We plan to use this technique in our future studies.

Al-Munajjed AA et al. J Biomed Mater Res B Appl Biomater 2009, 90(2), 584-591. doi: 10.1002/jbm.b.31320.

Vallecillo C et al. Polymers (Basel). 2021;13(16):2633. doi: 10.3390/polym13162633.

Xia Z et al. Acta Biomater 2013 9(7), 7308-19. doi: 10.1016/j.actbio.2013.03.038.

  1. Mention the reason for the following statement in line 334. “Both types of scaffolds, therefore, showed similar mechanical stability, which tended to be even higher in the UCS scaffolds than in the MCS scaffolds”. 

Reply: This statement originated from the Figure 5, where it is visible that the compression force, i.e. the force needed for 75% relative compression of the scaffolds, and the tension force, i.e. the force needed for stretching up to rupture of the scaffolds, were on an average higher in UCS than in MCS scaffolds, although these differences did not reach the level of statistical significance. We have interpreted these differences as a higher tendency of UCS to be mechanically more stable than MCS, i.e. more resistant to mechanical damage. A slightly worse mechanical stability of the MCS scaffolds can be explained by hydrolytic degradation of collagen in Tris-HCl buffer during mineralization, as discussed above in the point 4.

The statement in the manuscript has been more clearly formulated (Results, page 10, section “3.2. Mechanical stability of the scaffolds”).

  1. Why the sodium ion concentration in the UCS was more?

Reply: As explained in the manuscript, a lower content of Na in MCS than in UCS could be due to a higher release of sodium from MCS than from UCS. In MCS, sodium ions can, at least partly, substitute calcium ions in calcium phosphates like hydroxyapatite, but this substitution increases the instability, solubility and degradability of the calcium phosphates, followed by the release of sodium. Substitution of calcium with alkali metals (sodium, potassium, strontium) has been used for increasing the osteoconductivity of calcium phosphates, i.e. materials widely used for bone tissue engineering and regeneration. For example, potassium or sodium/strontium co-substituted calcium polyphosphate bioceramics increased the proliferation activity of rat osteoblast-like cells in vitro in comparison with non-substituted bioceramics (Song et al. 2011, reference 63 in the manuscript). Similarly, sodium-substituted hydroxyapatite exhibited markedly higher osteoconductivity than non-substituted hydroxyapatite, when implanted in calvarial defects of New Zealand white rabbits in vivo (Sang Cho et al. 2014, reference 64). These positive effects were attributed to a higher dissolution and degradation ability of the Na-substituted phosphates. In accordance with this, our MCS scaffolds also improved the cell growth directly on the scaffolds as well as in extracts from these scaffolds.

A deeper explanation of these problems associated with the sodium concentration in the mineralized collagen has been added to the Discussion (section “4.2. Positive effect of scaffold mineralization on cell colonization”, page 20).

Sang Cho J et al. J Biomed Mater Res B Appl Biomater 2014, 102(5), 1046-1062. doi: 10.1002/jbm.b.33087.

Song, W et al. J Biomed Mater Res B Appl Biomater 2011, 98(2), 255-262. doi: 10.1002/jbm.b.31847.

  1. Mention the reason why the occurrence of Calcium, Potassium, Magnesium, Phosphorus, and Sodium is found in the unmineralized scaffolds.

Reply: The occurrence of these elements in the unmineralized scaffolds can be explained by a potential contamination of the collagen sample, prepared from bovine skin, with interstitial fluid, which is a rich source of these elements (composition of SBF was inspired by the blood plasma, which is of a similar composition as the interstitial fluid). These elements are also contained in cells present in the bovine skin. The Ca2+, K+, Na+ and Mg2+ cations and phosphate anions can be strongly bound to the collagen molecule by electrostatic interactions, and therefore it can be difficult to remove them completely by the purification process of collagen.

This explanation has been added to the manuscript (Results, section “3.3. Elemental composition of the scaffolds”, page 10).

  1. Mention the reason why cells penetrated deeper in MCS as compared to UCS.

Reply: The penetration of cells inside the scaffolds is influenced by the amount and size of the pores within the scaffolds, by mechanical properties of the scaffolds, and by the chemical composition of the scaffolds. In MSC, the amount of pores with appropriate size (diameter) for the ingrowth of cells inside the scaffolds was lower than in UCS. The mechanical properties, i.e. the compression and tensile strength of the scaffolds, were similar, thus it can be supposed that both UCS and MCS scaffolds were able to provide a similar mechanical support for the cell adhesion, spreading and migration. Thus, the increased penetration of cells inside the MCS is most probably due to a favorable chemical composition of these scaffolds, particularly increased content of Ca, Mg and P. It is known that calcium is the simplest and most versatile second messenger mediating processes associated with the adhesion, migration, proliferation, survival, apoptosis, differentiation and phenotypic maturation of cells, including physiological healthy cells of various tissues and tumor cells, immature and differentiated cells, primary cell in vitro and cell lines, etc. (for a review, see Gopal et al. 2020, Lee et al. 2018, Resende et al. 2013, Patergnani et al. 2020). Ca2+ ions are essential for the adhesion function of integrins and other cell-matrix adhesion receptors, for the formation of focal adhesion plaques, for the assembly and integrity of actin cytoskeleton and other structures necessary for the cell attachment, spreading and migration (Gopal et al. 2020, Lee et al. 2018). Calcium is needed for the expression of genes involved in cell growth, for activation of tyrosine kinase receptors, cell cycle progression and other processes associated with cell proliferation (Resende et al. 2013, Patergnani et al. 2020). Magnesium and phosphorus also play positive roles in cell adhesion, migration and proliferation, i.e. key processes for cell penetration inside the scaffolds. Magnesium as a divalent cation can substitute calcium or act synergistically with this element in its activities supporting the cell adhesion and proliferation (Obata et al. 2019). Similarly, phosphorus also acts as an important activator of the cell proliferation. Elevated extracellular inorganic phosphate resulted in the induction of proliferation-associated genes, such as c-fos (Camalier et al. 2013). It is also generally known that phosphorylation of specific substrates by enzymes kinases, e.g. mitogen-activated protein kinases or cyclin-dependent kinases, is a key factor in induction of cell proliferation. Taken together, Ca, Mg and P and their compounds have a considerable osteoconductive effects, which lies in the stimulation of adhesion migration and proliferation of osteoblasts, and also an osteoinductive effect, i.e. ability to induce differentiation of immature cells, e.g. stem cells, towards osteoblasts (Jeong et al. 2019, Wang et al. 2019).

These problems are discussed in the Discussion, section “4.2. Positive effect of scaffold mineralization on cell colonization” (page 19), and this discussion has been further enriched with some remarks mentioned here.

Camalier CE et al. J Cell Physiol 2013, 228(7), 1536-1550. doi: 10.1002/jcp.24312.

Gopal S et al. Adv Exp Med Biol. 2020; 1131:1079-1102. doi: 10.1007/978-3-030-12457-1_43.

Jeong J et al. Biomater Res 2019, 23:4. doi: 10.1186/s40824-018-0149-3.

Lee MN et al. Exp Mol Med 2018, 50(11), 1-16. doi: 10.1038/s12276-018-0170-6.

Obata A et al. J Biomed Mater Res A 2019, 107(5), 1042-1051. doi: 10.1002/jbm.a.36623.

Patergnani S et al. Int J Mol Sci. 2020; 21(21):8323. doi: 10.3390/ijms21218323.

Resende RR et al. Cell Commun Signal. 2013; 11(1):14. doi: 10.1186/1478-811X-11-14.

Wang, S et al. J Mech Behav Biomed Mater 2019, 94, 42-50. doi: 10.1016/j.jmbbm.2019.02.026.

 Other remarks: New Figure 7 and new references have been added into the manuscript. The figures and the references have been reorganized and renumbered.

Reviewer 2 Report

In this manuscript by Lucie Bacakova et al., the authors studied the effect of the mineralized collagen scaffold for the bone cell growth. They measured the spatial structure, the chemical composition, the compression and tensile responses, the cell spreading, and the immune activity of the cell. In order to evaluate the effect of the mineralization, they showed these physical properties of the mineralized scaffold (MCS) and unmineralized one (UCS) which show high and low densities of chemical components (Ca, Mg and P), respectively.

Concerning result for the spatial structure, the MCS includes mainly small-size pores, while the UCS shows the normal distribution of the pore size. Concerning the result for the mechanical properties, the MCS and the UCS show the similar stiffnesses for the compression and the extension. In the early stage of the cell spreading, the number of cells in the MCS is larger than that in UCS. The result for the immune activity shows higher activity of MCS rather than that of UCS.

Then, the author concluded that the MCS is appropriate for the bone tissue engineering.

The result in this manuscript seems to be important in the context of the engineering research. On the other hand, the origin of the results on the MCS are hardly understood for the reviewer. Moreover, some points are difficult for the reviewer. The reviewer cannot agree with accept unless the manuscript is revised. The authors are requested to consider the following comments on revision of the manuscript.

L13:

What is the simulated body fluid? Is this the solution under the physiological condition?

L18:

It is difficult for the reviewer to understand “cell penetration”. The author should, in more detail, explain it in the main text.

L22:

MSC should be MCS.

L48:

The “weak mechanical properties” is difficult for the reviewer to understand. What properties are focused on? Compression strength, stress-strain curve or something else? Moreover, in this context, what is the “strong mechanical properties”?

L100:

The reviewer cannot find what theoretical or experimental results motivated the study on “porous” collagen. The authors should mention how important the “porous” collagen scaffold is studied.

L331:

 The force needed for stretching up to the rapture of the scaffold are measured. If the scaffold is spatially anisotropic, that force depends on its direction. However, the force direction is not mentioned as well as the anisotropy. The author should show the force direction and the spatial anisotropy which can be derived from the confocal microscopy images.

Section 4 Discussion:

The author should refer to Figure XX at each paragraph when the result is mentioned.

For example, in the first paragraph in section 4.1, Figure of the result is not referred, although the result in this present study and the previous studies are compared.

Section 4.1:

Although the correlation between the size of pore and the cell size are pointed out based on the experimental results, its mechanism is not mentioned. The author should show it (or hypothesis of the origin).

L472:

The reviewer hardly understands “small pores may result in a large surface.” Why? The author should, in more detail, explain that.

Section 4.2:

Although the correlation between the scaffold mineralization and the cell colonization is discussed, its mechanism is not mentioned. The authors should explain the origin of the correlation. For example, the relationship between the pore size distribution and the mineralization seems to be important.

Section 4.4:

The mechanism of the immunological acceptance of the scaffold is also required.

Section 4:

What characteristics of the collagen (polymer) leads to the experimental result in the present research?

Author Response

The authors thank Reviewer 2 for his (her) valuable comments. All these comments have been carefully considered, and the paper has been modified according to these suggestions (the changes are highlighted in yellow color in the enclosed manuscript).

L13:

What is the simulated body fluid? Is this the solution under the physiological condition?

Reply: The simulated body fluid (SBF) is artificially-prepared solution of ions in a buffer (TRIS or HEPES) with similar concentration of these ions and pH as in the blood plasma. These ions include Na+, K+, Mg2+, Ca2+, Cl, HCO3, HPO42− and SO42−. The SBF was first introduced by Kokubo et al. (1990) in order to reproduce structure changes of the surface of bioactive glass ceramics after implantation in vivo - i.e. the formation of bone-like apatite on the biomaterial surface. Such, this solution imitates physiological conditions, and as an alternative, also cell culture media (like DMEM used for cell cultivation on our study), can be applied. 

The ion species contained in the SBF have been mentioned in Materials and Methods (section “2.1.2. Biomimetic mineralization of collagen scaffolds with SBF”, pages 3-4).

Kokubo T et al. J Biomed Mater Res. 1990;24(6):721-34. doi: 10.1002/jbm.820240607. 

L18:

It is difficult for the reviewer to understand “cell penetration”. The author should, in more detail, explain it in the main text.

Reply: The cell penetration into a scaffold is a term used to describe the colonization of the scaffolds with cells, i.e. first the cell adhesion to the surface of the scaffolds, then the migration of cells into the scaffolds and their proliferation within the scaffolds. This term can be replaced with the words “cell ingrowth”, “cell immigration”, “cell infiltration”, “cell colonization” etc., which we did throughout the entire manuscript.

L22:

MSC should be MCS.

Reply: we have corrected this typographical error.

L48:

The “weak mechanical properties” is difficult for the reviewer to understand. What properties are focused on? Compression strength, stress-strain curve or something else? Moreover, in this context, what is the “strong mechanical properties”?

Reply: Biomaterials designed for tissue replacements and tissue engineering are most frequently characterized from the point of view of the stresses and strains in a material that result from applied loads and displacements. The most fundamental mechanical properties of materials are obtained from a stress-strain curve. These properties include the elastic modulus (E), yield strength and strain, ultimate strength and strain, fracture strength, strain-to-failure, modulus of resilience and modulus of toughness (for a review, see Roeder et al. 2013). Various biomaterial, such as ceramics, metals, natural and synthetic polymers, and various composites of these main classes of biomaterials are most frequently compared on the basis of the elastic modulus, i.e. Young´s modulus (E), which is an indicator of the material stiffness. This modulus is relatively low in polymeric materials (usually in the order of kPA, MPa or units of GPa), but relatively high in ceramics and metals (tens or hundreds of GPa; for a review, see Roeder et al. 2013; Reddy et al. 2021).

Young´s modulus of bovine tendon collagen was relatively high, ranging from 1.0 GPa to 3.9 GPa (Yang et al. 2009), and for individual type 1 collagen fibrils from rat tail, this modulus was found to range even from 5 GPa to 11.5 GPa (Wenger et al. 2007). However, Young´s modulus of porous collagen scaffolds used for bone tissue engineering was generally much lower - for example, Young´s modulus of collagen scaffolds prepared by freeze-drying like in our study was only 1-10 kPa depending on the hydrated or dry state (Varley et al. 2016). In contrast, the Young´s modulus of human bone tissue has been reported to range approx. from 2 to 22 GPa depending on the evaluation method (bending tests, compression tests, ultrasonic testing finite element analysis, nanoindentation), anatomical site (femur, tibia, iliac bone, vertebrae), dry or wet condition, or age and gender of the donor (for a review, see Wu et al. 2018). The “weak mechanical properties” mentioned in our study therefore refer to the properties derived from a stress-strain curve, particularly Young´s modulus, and the text in the manuscript has been clarified in this sense. In addition, these properties led us to reinforce the collagen scaffolds in our study, preferably with a mineral material, which resembles the physiological inorganic component of the bone matrix.

These aspects have been briefly mentioned in the Introduction, page 2.

Reddy MSB et al. Polymers (Basel). 2021;13(7):1105. doi: 10.3390/polym13071105.

Roeder RK. In: Characterization of Biomaterials, Eds: Bandyopadhyay A, Bose S. Elsevier Inc. 2013, chapter 3, pp. 49-104.  ISBN 978-0-12-415800-9, doi: 10.1016/C2011-0-04481-2

Varley MC et al. Acta Biomater. 2016;33:166-75. doi: 10.1016/j.actbio.2016.01.041.

Wenger MP et al. Biophys J. 2007;93(4):1255-63. doi: 10.1529/biophysj.106.103192.

Wu D et al. Acta Biomater. 2018;78:1-12. doi: 10.1016/j.actbio.2018.08.001.

Yang L et al. Biophys J. 2008;94(6):2204-11. doi: 10.1529/biophysj.107.111013.

L100:

The reviewer cannot find what theoretical or experimental results motivated the study on “porous” collagen. The authors should mention how important the “porous” collagen scaffold is studied.

Reply: Porous scaffolds are an important type of three-dimensional (3D) materials used for tissue engineering. There are two basic types of 3D porous scaffolds: sponge-like scaffolds and fibrous scaffolds. Sponge-like structure can be created e.g. by freeze drying (Varley et al. 2016, our present study) or by salt leaching (Pamula et al. 2009). Fibrous scaffolds can be created by spinning techniques, such as electrospinning or centrifugal spinning (Parizek et al. 2012, Novotna et al. 2014, Mahalingam et al. 2014). Techniques used for creating porous sponge-like and fibrous scaffolds can be also combined - for example, pores in fibrous scaffolds, which are often too small for proper formation of bone tissue, can be enlarged by so-called “sacrificial particles”, made e.g. from NaCl, sucrose or water-soluble polymers (Wang et al. 2013).

The porous structure of the scaffolds is especially advantageous for bone tissue engineering, because it resembles the physiological architecture of the native bone tissue. This tissue is of two main types: cortical (compact) bone, which is composed of osteons (i.e. Haversian systems), consisting of several concentric layers of osteoblasts and osteocytes around a central canal with blood vessel supply, and cancellous bone, also called trabecular or spongy bone, which is in fact a foam-like network with open pores. The pores in a scaffold designed for bone tissue engineering should be sufficiently large (usually at least 100 um in diameter) in order to allow accommodation of osteoblasts, production and deposition of extracellular matrix by these cells, formation of osteon-like systems with vascularization (for a review, see Karageorgiou and Kaplan 2005, Kołodziejska et al. 2020). It is also desirable if the pores are interconnected, which allows them to process a significant amount of water and facilitates the flow of fluids (Granito et al. 2016), and if the scaffolds have a hierarchically-organized macro-micro-nano- structure, similarly as in the native bone tissue (for a review, see Bacakova et al. 2016, Kołodziejska et al. 2020).

The advantages of porous scaffolds for bone tissue engineering, including porous collagen scaffolds, have been briefly explained in the Introduction of the manuscript (page 2).

Bacakova L et al. In: Nanobiomaterials in Hard Tissue Engineering. Applications of Nanobiomaterials. Volume 4, Ed. A. M. Grumezescu, Elsevier Inc., William Andrew Publishing, Oxford, Cambridge; Chapter 4, pp. 103-153, ISBN 978-0-323-42862-0; 2016

Granito RN et al. J Biomed Mater Res B Appl Biomater. 2017;105(6):1717-1727. doi: 10.1002/jbm.b.33706.

Karageorgiou V, Kaplan D. Biomaterials 2005, 26(27), 5474-5491. doi: 10.1016/j.biomaterials.2005.02.002.

Kołodziejska B et al. Int J Mol Sci. 2021;22(12):6564. doi: 10.3390/ijms22126564.

Mahalingam S et al. Carbohydr Polym. 2014;114:279-287. doi: 10.1016/j.carbpol.2014.08.007.

Novotna K et al. J Biomed Mater Res A 2014, 102(11), 3918-3930. doi: 10.1002/jbm.a.35061.

Pamula E et al. J Biomed Mater Res A 2009, 89(2) 432-443. doi: 10.1002/jbm.a.31977.

Parizek M et al. Int J Nanomedicine 2012, 7, 1931-1951. doi: 10.2147/IJN.S26665.

Varley MC et al. Acta Biomater. 2016;33:166-75. doi: 10.1016/j.actbio.2016.01.041.

Wang K et al. J Biomed Mater Res A. 2013;101(12):3474-81. doi: 10.1002/jbm.a.34656.

L331:

 The force needed for stretching up to the rapture of the scaffold are measured. If the scaffold is spatially anisotropic, that force depends on its direction. However, the force direction is not mentioned as well as the anisotropy. The author should show the force direction and the spatial anisotropy which can be derived from the confocal microscopy images.

Reply: The images indicate that the samples are isotropic. We did not observe any preferential directionality within the scaffolds. This was highly expected as the strips are prepared from the suspension, therefore the porous structure used for the strips has no reference for initial directionality. The strips were always cut in a non-reference system. If there was any mechanical anisotropy, we would experience mixture of two populations depending on the directionality of the fibers at the test. Furthermore, the samples are returning similar results in both tensile and compression test. As in this case there are orthogonal to each, unlike the classical tensile/compression test, they prove the mechanical anisotropy of the samples, which is, at the end, quite typical for materials with high porosity. The mechanical test was not accompanied with the confocal microscopy and while using TA-XT2i, this combination is not an option, however, both the confocal microscopy and mechanical testing proves the mechanical and structural isotropy of the scaffolds.

Section 4 Discussion:

The author should refer to Figure XX at each paragraph when the result is mentioned.

For example, in the first paragraph in section 4.1, Figure of the result is not referred, although the result in this present study and the previous studies are compared.

Reply: The references to Figures have been added to each paragraph of the Discussion.

Section 4.1:

Although the correlation between the size of pore and the cell size are pointed out based on the experimental results, its mechanism is not mentioned. The author should show it (or hypothesis of the origin).

Reply: The cells of human or animal tissues, including osteoblasts or other osteogenic cells (e.g. mesenchymal stem cells) have around 15 µm in diameter, when they acquire round shape in suspension, and spread over tens of micrometers, when seeded on a material in vitro. However, the diameter of pores within the scaffolds designed for bone tissue engineering should be much larger, at least 100 µm in diameter. The reason is that for proper bone tissue formation, the osteogenic cells need a similar environment as in the natural bone tissue. In cortical bone, the diameter of Haversian systems is minimally 100 µm (for a review, see Karageorgiou and Kaplan 2005, Kołodziejska et al. 2020), and the size of the pores in cancellous bone is even larger, having 100-600 µm (Doktor et al. 2011). This pore size is required for the adhesion, migration and proliferation of osteogenic cells, formation of osteon-like structures with central blood supply, production of mineralized bone matrix. As described in the section 4.1. of the Discussion, if the pores are smaller, they are colonized either with osteoblasts without bone matrix mineralization or with other cell types, particularly fibroblasts (Karageorgiou and Kaplan 2005). Too small pores, e.g. 40 µm or less in diameter, can even be covered by cells in the manner of a “lid”, which prevents colonization of the inner part of the scaffolds with any cells (Pamula et al. 2008). However, very large pores, i.e. of diameter 1000 µm and more. are not suitable for the bone tissue regeneration, too. When osteoblasts adhere on the walls of such pores, they “sense” such a substrate as two-dimensional rather than three-dimensional, behave as in conventional 2D cultures and do not form appropriate bone tissue (Frosch et al. 2002, Murphy and O'Brien 2010, Xia et al. 2013). Thus, the optimum pore size in scaffolds for bone tissue regeneration are in the middle of the range of 100-1000 µm, being approx. from 300 to 800 µm in various studies performed on various materials (Pamula et al. 2009, Murphy and O'Brien 2010, Xia et al. 2013).

The most important of these remarks have been added to Discussion (pages 17-18).

Doktor T et al. Proceedings of the 17th International Conference Engineering Mechanics; Svratka, Czech Republic. 9–12 May 2011.

Frosch et al. Cells Tissues Organs 2002; 170(4), 214-227. doi: 10.1159/000047925.

Karageorgiou V, Kaplan D. Biomaterials 2005, 26(27), 5474-5491. doi: 10.1016/j.biomaterials.2005.02.002.

Kołodziejska B et al. Int J Mol Sci. 2021;22(12):6564. doi: 10.3390/ijms22126564.

Murphy CM, O'Brien FJ. Cell Adh Migr. 2010;4(3):377-81. doi: 10.4161/cam.4.3.11747.

Pamula E et al. J Mater Sci Mater Med. 2008;19(1):425-35. doi: 10.1007/s10856-007-3001-1.

Pamula E et al. J Biomed Mater Res A 2009, 89(2) 432-443. doi: 10.1002/jbm.a.31977.

Xia Z et al. Acta Biomater. 2013;9(7):7308-19. doi: 10.1016/j.actbio.2013.03.038.

L472:

The reviewer hardly understands “small pores may result in a large surface.” Why? The author should, in more detail, explain that.

Reply: It is generally known that if sponge-like scaffolds of the same porosity (i.e. the percentage of void space in the material) differ in the pore diameter, the smaller pores are more numerous and result in a larger inner specific surface of the foam scaffolds. In other words, scaffolds for tissue engineering with smaller and more numerous pores possess (at least theoretically) a larger area available for cell colonization than scaffolds with larger and less numerous pores. However, this statement may come into conflict with the finding on the optimum pore size for cell ingrowth described above. If the pores are too small for immigration of cells inside the scaffolds, the cells cannot use the inner scaffold area for their attachment growth. However, they can use the outer surface area of these scaffolds, which is more appropriate for the cell attachment and spreading than the surface with larger pores. As is obvious from Figures 2 a 3 in our manuscript, on the surface with smaller pore diameter, the cells can find more sites for their attachment and can bridge the void spaces more easily than on surfaces with a large pore diameter. Thus, it can be concluded that on materials with a larger pore diameter, the cells get better inside the material, while on the material with smaller pores, they can attach, proliferate can reach a higher population density on the surface of the material. In ideal scaffolds, both of these aspect must be balanced (Murphy and O'Brien 2010, Xia et al. 2013).

This problem has been better explained in the Discussion (page 18).

Murphy CM, O'Brien FJ. Cell Adh Migr. 2010;4(3):377-81. doi: 10.4161/cam.4.3.11747.

Xia Z et al. Acta Biomater 2013 9(7), 7308-19. doi: 10.1016/j.actbio.2013.03.038.

Section 4.2:

Although the correlation between the scaffold mineralization and the cell colonization is discussed, its mechanism is not mentioned. The authors should explain the origin of the correlation. For example, the relationship between the pore size distribution and the mineralization seems to be important.

Reply: The mineralization of the scaffolds can support the cell colonization by two basic mechanisms: mechanical and biochemical signaling. It the mineralization increases the stiffness of the scaffolds, these scaffolds provide a stronger support for the cell adhesion and migration (i.e., the material can better resist the traction forces generated by cells during spreading and migration, and does not collapse under the cells), and also promote osteogenic differentiation of cells. It is known that the stiffness of the cell adhesion substrate can determine the direction of cell differentiation. For example, on soft polyacrylamide gels bound with collagen (modulus of elasticity E = 0.1-1 kPa), bone marrow mesenchymal stem cells differentiated towards neuronal cells, while on stiffer gels (E = 8-17 kPa), they differentiated towards muscle cells, and on the stiffest gels (E = 25-40 kPa), they differentiated towards osteoblast (Engler et al. 2006, reference 54 in the manuscript). However, in our collagen scaffolds, where the compression and tensile strength were not improved by mineralization, the increase in cell colonization can be attributed rather to biochemical signaling from the mineral component of the scaffolds. It is known that Ca, Mg and P ions are potent stimulators of the cell adhesion, migration and proliferation by biochemical pathways - for a more detailed explanation, please, see the response to the point 8 raised by Referee 1. Biochemical signaling as a main cause of the increased growth of cells on mineralized scaffolds is further supported by the fact that increased cell proliferation also occurred in extracts from these scaffolds, i.e. under conditions when the mechanical properties of the scaffolds played no role.

The mechanism by which the mineralization of the scaffolds improves the cell colonization are discussed in more details in the section 4.2. Positive effect of scaffold mineralization on cell colonization” of the Discussion (page 19).

An interesting issue is that although the cell colonization was improved in mineralized scaffolds, these scaffolds contained a lower amount of pores with optimum size for the cell ingrowth. In the literature, the mineralization of the scaffolds, i.e. an increased content of minerals inside the scaffolds, had often negative effect on the pore size. As is explained in the Discussion (section “4.3 Extrafibrillar versus intrafibrillar mineralization of collagen”, page 21), this negative effect is typical especially for such scaffolds which were mineralized extrafibrillary, e.g. by a simple immersion in SBF or related ionic solution, similarly as in the present study, or by admixing a mineral component into a collagen matrix. In these cases, the mineral component is located on the surface of collagen fibrils, and can restrict considerably the size of the pores. However, when the collagen is mineralized intrafibrillary, i.e. within the gap zone of collagen fibrils, similarly as in the native bone tissue, the restriction of the pore size is less apparent. Thus in our future studies, we plan to apply the intrafibrillar collagen mineralization by a Polymer Induced Liquid Precursor (PILP) process (please, also see the response to Referee 1, point 4, and the section 4.3 in the Discussion - page 21).

Engler, A.J.; Sen, S.; Sweeney, H.L., Discher, D.E. Matrix elasticity directs stem cell lineage specification. Cell 2006, 126(4) 677-689. doi: 10.1016/j.cell.2006.06.044.

Section 4.4:

The mechanism of the immunological acceptance of the scaffold is also required.

Reply: Collagen scaffolds for tissue engineering are usually made of xenogeneous collagen, e.g. of bovine or porcine origin. Although the bovine or porcine collagen is approved for biomedical use, including implantation or injection into patients, it is recognized as a foreign material and can evoke inflammatory reaction in humans (Alijotas-Reig et al. 2013) as well as in laboratory animals, such as mice (Rücker et al. 2006) or rats (Panda et al. 2013). This reaction is based on non-specific (innate) immunity, which is mediated by leucocytes, monocytes and macrophages, and specific immunity (i.e. acquired during the organism's lifetime), mediated by B and T lymphocytes. The cells of non-specific immunity eliminate the foreign material by phagocytosis and degradation by lysosomal enzymes. At the same time, they alert the cells of the specific immunity (lymphocytes) - e.g. by production of pro-inflammatory cytokines, such as TNF-alpha and interleukins-1. The cells of the specific immunity then produce antibodies against the foreign antigens (B lymphocytes), destroy the antigens by production of proteases (T lymphocytes), and also create immunological memory (both B and T lymphocytes) (for a review, see Janeway et al. 2001). Interestingly, since xenogeneous or allogeneous collagen molecules are similar to patient´s own collagen molecules, the specific immune reaction could be directed not only to the foreign collagen, but also against the patient's own collagen, and due to the immunological memory, this reaction could last even after elimination of foreign collagen from the patient´s body, which is one of the mechanisms of autoimmune diseases. On the other hand, pure collagen, even xenogeneous, is much less immunogenic than tissue transplants containing cells. It has been reported that after decellularization, a tissue loses up to 90% of its immunogenicity (Boccafoschi et al. 2017).

In our present study, only a part of non-specific immunity mediated by macrophages was investigated. We found that collagen can evoke immune activation of these cells, manifested by production of TNF-alpha, and this activation can be further enhanced on mineralized scaffolds. It is known that calcium is an important mediator of inflammatory reaction, especially when it enters in direct contact with cells, e.g. in case of extrafibrillary-mineralized scaffolds. Therefore, in our future studies, we plan to alleviate the inflammatory reaction, evoked by mineralized collagen scaffolds, by intrafibrillar mineralization of these scaffolds. In a study by Sun et al. (2016) cited in the Discussion (ref. 39), intrafibrillary-mineralized collagen attracted predominantly anti-inflammatory macrophages of M2 phenotype, which are associated with the resolution phase of inflammation and with the repair of damaged tissues, while extrafibrillary mineralized collagen scaffolds attracted predominantly pro-inflammatory macrophages of M1 phenotype. We therefore expect that the intrafibrillary mineralization would enhance the immunological acceptance of the scaffolds. Intrafibrillary-mineralized collagen would induce only a mild inflammatory reaction, desirable for gradual removal of the implanted material and its replacement with a newly formed regenerated tissue.

The mechanism of the immune acceptance/rejection of the collagen scaffolds, including the possibility of influencing this phenomenon, has been briefly described in the Discussion (section 4.4. Immunological acceptance of the scaffolds, pages 21-22).

Alijotas-Reig J et al. Semin Arthritis Rheum. 2013;43(2):241-58. doi: 10.1016/j.semarthrit.2013.02.001.

Boccafoschi F et al. J Tissue Eng Regen Med. 2017;11(5):1648-1657.

Janeway CA Jr et al. Immunobiology, 5th edition. The Immune System in Health and Disease. New York: Garland Science; 2001. ISBN-10: 0-8153-3642-X

Panda NN et al. J Biomater Sci Polym Ed. 2013, 24(18), 2031-2044. doi: 10.1080/09205063.2013.822247.

Rücker M et al. Biomaterials 2006, 27(29), 5027-5038. doi: 10.1016/j.biomaterials.2006.05.033.

Sun Y et al. J Biomed Nanotechnol 2016, 12(11), 2029-2040. doi: 10.1166/jbn.2016.2296.

Section 4:

What characteristics of the collagen (polymer) leads to the experimental result in the present research?

Reply: collagen is a biopolymer which is the most abundant structural component of the extracellular matrix of most tissues, particularly connective tissue including bone. In mammals, this protein makes up 25% to 35% of the whole-body protein content. Collagen is also an important component of the bone tissue. About 30-40 % of the bone tissue is composed of collagen, particularly type I collagen. Type I collagen is the major protein component of the bone extracellular matrix, accounting for up to 90% of the organic matrix. Together with collagen types II, III, V and XI, type I collagen belongs to fibril-forming collagens, characterized by their primary, secondary and tertiary structure. As a protein, collagen is composed of long chains of amino acids, and the identity and sequence of these amino acids defines the primary structure of collagen. The most abundant amino acids in the collagen molecule are glycine, proline, alanine, hydroxyproline. The amino acid chains are arranged into helices, which represent the secondary structure of collagen. Three interconnected helical chains then form a triple helix, which is the tertiary structure of collagen (for a review, see (Henriksen and Karsdal 2016, Maher et al. 2021, Yu and Wei 2021).

Specific amino acid sequences within the collagen molecules serve as ligands for integrin cell adhesion receptors, thus collagen enables direct cell attachment without need of adsorption of cell adhesion-mediating molecules, e.g. vitronectin of fibronectin from the culture medium, which is required in other polymeric materials, e.g. synthetic polymers or some nature-derived polymers of non-mammalian origin, such as cellulose or chitosan. These sequences include GFOGER in physiological native triple-helical collagen (Knight et al. 2000), and DGEA and RGD in modified collagen, like heat-denatured collagen and gelatin (Yamamoto et al. 1995). In addition, collagen is degradable and reorganizable by cells, and can be easily replaced with their own newly synthesized ECM. Finally, collagen provides the structural matrix upon which mineralization of bone occurs.

It is therefore reasonable to use collagen as a material for constructing scaffolds for bone tissue engineering. Supportive effect of collagen on the growth of human bone-derived cells was proved in the present study, including a further improvement of the cell growth by collagen mineralization. Some of these remarks have been added to the Introduction (page 2).

Henriksen K, Karsdal MA. Type I Collagen. In: Biochemistry of Collagens, Laminins and Elastin, Structure, Function and Biomarkers. Ed. Karsdal MA, Academic Press, Elsevier Inc. 2016, chapter 1, pp, 1-11, ISBN 978-0-12-809847-9, doi: 10.1016/C2015-0-05547-2.

Maher M et al. Acta Biomater 2021, S1742-7061(21)00409-8. doi: 10.1016/j.actbio.2021.06.035.

Yu L,Wei M. Int J Mol Sci 2021, 22(2), 944. doi: 10.3390/ijms22020944.

Knight CG et al. J Biol Chem. 2000;275(1):35-40. doi: 10.1074/jbc.275.1.35.

Yamamoto M et al. Exp Cell Res. 1995;219(1):249-56. doi: 10.1006/excr.1995.1225.

Other remarks: New Figure 7 and new references have been added into the manuscript. The figures and the references have been reorganized and renumbered.

Round 2

Reviewer 2 Report

In this revised manuscript by Lucie Bacakova et al., the authors nicely revised the manuscript so that it is easier for the reviewer (and some readers) to understand it. However, the reviewer seems that the authors should justify or revise the just one point. The reviewer recommends the publications after the revision.

Figure 5:

 The authors answer to the reviewer’s question about the anisotropy and the force direction in which the compression strength and the tensile strength are measured. According to authors answer, the scaffold seems isotropic texture in the confocal microscopic image. If the system is isotropy, the response to the external force is, in general, also isotropy. Therefore, the compression and tensile strengths should show the same trends as the authors showed in the manuscript. The reviewer recommends that it is enough showing either of strength or tensile strengths in Figure 5.

Author Response

Answer: According to the Reviewer’s comment, we have shown only tensile strength in Figure 5, and the values of compression strength have been mentioned only in the text (section 3.2. Mechanical stability of the scaffolds) in order to show the same trend of these values as in the compression strength.

The new changes, which have been made in Materials and Methods, Results and Figure 5, are highlighted in green.
